# Guanine nucleotide biosynthesis blockade impairs MLL complex formation and sensitizes leukemias to menin inhibition

Xiangguo Shi[1,2,3,4] ✉, Minhua Li[5], Zian Liu[6], Jonathan Tiessen[7], Yuan Li[8], Jing Zhou[5], Yudan Zhu[4], Swetha Mahesula[9], Qing Ding[9], Lin Tan[10], Mengdie Feng [4], Yuki Kageyama [4], Yusuke Hara[4], Jacob J. Tao [5], Xuan Luo[1], Kathryn A. Patras [11], Philip L. Lorenzi[10], Suming Huang [1,3], Alexandra M. Stevens[12], Koichi Takahashi [13,14], Ghayas C. Issa [13], Md. Abul Hassan Samee [6], Michalis Agathocleous [9] & Daisuke Nakada [4,5,7] ✉

Targeting the dependency of *MLL*-rearranged (*MLL*r) leukemias on menin with small molecule inhibitors has opened new therapeutic strategies for these poor-prognosis diseases. However, the rapid development of menin inhibitor resistance calls for combinatory strategies to improve responses and prevent resistance. Here we show that leukemia stem cells (LSCs) of *MLL*r acute myeloid leukemia (AML) exhibit enhanced guanine nucleotide biosynthesis, the inhibition of which leads to myeloid differentiation and sensitization to menin inhibitors. Mechanistically, targeting inosine monophosphate dehydrogenase 2 (IMPDH2) reduces guanine nucleotides and rRNA transcription, leading to reduced protein expression of LEDGF and menin. Consequently, the formation and chromatin binding of the MLL-fusion complex is impaired, reducing the expression of MLL target genes. Inhibition of guanine nucleotide biosynthesis or rRNA transcription further suppresses *MLL*r AML when combined with a menin inhibitor. Our findings underscore the requirement of guanine nucleotide biosynthesis in maintaining the function of the LEDGF/menin/MLL-fusion complex and provide a rationale to target guanine nucleotide biosynthesis to sensitize *MLL*r leukemias to menin inhibitors.

Acute myeloid leukemia (AML) is a heterogeneous hematologic malignancy that originates from the transformation of hematopoietic stem and progenitor cells (HSPCs) and is the most common leukemia in adults. Despite significant advancements in our understanding of AML pathogenesis and the development of new treatments, the five-year survival rate for AML patients, especially those aged over 75 years, remains dismal. These persistent challenges in effectively treating AML emphasize the urgent need for innovative and targeted therapeutic strategies to cure this devastating disease.

Chromosomal rearrangements involving chromosome 11q23 are among the most frequent cytogenetic aberrations in pediatric AML and acute lymphoblastic leukemia and are also found in 5-10% of adult acute leukemias[1,2]. 11q23 rearrangement involves the *KMT2A/MLL* gene and produces MLL-fusion proteins with over 80 translocation partners, with AF9 being one of the most common fusion partners in adult and pediatric AML[3]. The oncogenic MLL-fusion proteins interact and recruit the super elongation complex (SEC) to the MLL target genes and drive transcriptional elongation of its target genes such as the

HOX genes[1,4,5]. The recruitment of MLL and MLL-fusion proteins to the target genes is in turn regulated by LEDGF, which interacts with the N-terminal of MLL via an adaptor protein menin[6,7]. Biophysical characterization of the menin-MLL interaction led to the development of small molecule inhibitors of menin-MLL interaction[8]. Recent clinical trials with menin inhibitors show promising results against *MLL*-rearranged (*MLL*r) AML, although resistance frequently occurs, rendering the menin-MLL interaction insensitive to the drug[9,10]. New strategies to synergistically suppress target recognition by MLL-fusion by interfering with menin or LEDGF may improve AML therapy.

An emerging avenue of investigation in AML concerns the abnormal metabolic processes within the leukemia cells. This strategy acknowledges leukemia cells' unique metabolic demands relative to normal hematopoietic cells, leveraging these distinctions to eliminate AML cells while preserving normal hematopoiesis. Notable metabolic alterations observed in AML cells, particularly leukemia stem cells (LSCs) that are largely responsible for chemoresistance and relapse[11], include increased reliance on oxidative phosphorylation (OXPHOS), amino acid metabolism, fatty acid metabolism, and redox maintenance[12–17]. Notably, targeting the OXPHOS with venetoclax, a B-cell lymphoma 2 (BCL2) inhibitor, achieves remarkable efficacy in cases of de novo AML, but has limitations in treating relapsed/refractory or monocytic AML[14,18]. These findings underscore the critical importance of gaining a comprehensive understanding of metabolic regulations in AML, particularly within the context of LSCs, to develop effective therapeutic interventions.

Here, we find that the guanine nucleotide biosynthesis pathway is highly activated in LSCs of *MLL*r AML. Pharmacological inhibition of IMPDH drives differentiation of immature AML cells accompanied by reduced expression of key leukemogenic genes downstream of the MLL-fusion oncoprotein. Mechanistically, IMPDH inhibition reduces the expression, complex formation, and chromatin binding of the LEDGF-menin-MLL complex and renders *MLL*r AML cells highly sensitive to menin inhibitors. Our study thus unveils the mechanistic basis in which inhibition of the guanine nucleotide biosynthesis renders AML susceptible to menin inhibition.

## Results

### Purine biosynthetic metabolites are enriched in LSCs

To uncover the metabolic landscape of the immature fraction of AML, we compared the metabolic profiles of leukemic granulocyte-macrophage progenitors (L-GMPs)[19] and bulk AML cells derived from an MLL-AF9-driven murine AML model, with normal GMPs and whole bone marrow (WBM) cells from C57BL/6 mice using liquid chromatography-mass spectrometry[20] (Fig. 1a). L-GMPs have been shown to possess LSC activity in the MLL-AF9-induced AML model[19], and will be referred to as LSCs in this study. We detected 128 metabolites within several metabolic pathways, including glycolysis, amino acid metabolism, and nucleotide biosynthesis (Supplementary Data 1). Samples from each population clustered together, indicating that each population had its own metabolite profile (Supplementary Fig. 1a–c). By comparing the enrichment of these metabolites in the four populations, we found that LSCs and bulk AML were different from each other but were closer to each other than to the non-leukemic populations (Fig. 1b and Supplementary Fig. 1a, b). The metabolite profiles of the two leukemic populations were more similar to normal GMP than to WBM cells, suggesting the existence of a myeloid metabolite signature (Supplementary Fig. 1c). Consistent with previous findings that LSCs demonstrate elevated levels of amino acids[13], we found that MLL-AF9-driven LSCs exhibit a higher abundance of metabolites enriched in gene ontology (GO) terms such as aminoacyl-tRNA biosynthesis, arginine metabolism, and glycine, serine, and threonine metabolism when compared to GMP and bulk AML cells (Supplementary Fig. 1d–g). Moreover, although GMPs displayed the highest

levels of glycolysis intermediates and lower levels of metabolites associated with the TCA cycle, LSCs exhibit elevated levels of TCA cycle intermediates such as citrate/isocitrate and aconitate (Supplementary Fig. 1h, i). Importantly, among all the metabolites enriched in LSCs, we found an increased abundance of metabolites in the purine biosynthetic pathway including guanosine, succinyl-5-aminoimidazole-4-carboxamide-1-ribose-5′-phosphate (SAICAR), and allantoin (Fig. 1c–e and Supplementary Fig. 1j), although the detection levels of some of these metabolites were variable. These data suggest that the immature state of AML is associated with an active purine biosynthesis pathway.

To assess the metabolic flux through purine biosynthesis in LSCs, we performed isotope tracing experiments by treating LSCs and non-LSCs with $^{13}C_6$-glucose or amide-$^{15}N$-glutamine for 1 or 4 h, and analyzed the incorporation of $^{13}C$ and $^{15}N$ into purine metabolites using ion chromatography-mass spectrometry (IC-MS)[21,22] (Supplementary Fig. 1k). Consistent with the increased levels of purine metabolites in LSCs observed through targeted metabolomics, we noted higher $^{13}C$ isotopic enrichment in IMP, guanosine mono, di, and trinucleotides, AMP/ATP/ADP in LSCs compared to non-LSCs (Fig. 1f). More specifically, LSCs exhibited a greater fractional abundance of the $m+5$, $m+6$, and $m+7$ carbon forms of purine nucleotides (Fig. 1g–l). Similar findings were observed in LSCs treated with amide-$^{15}N$-glutamine, which provides two nitrogen atoms to IMP and one additional nitrogen to guanosine mononucleotide. LSCs showed higher $^{15}N$ isotopic enrichment over time, with increased $m+1$ and $m+2$ nitrogen forms of purine nucleotides than non-LSCs (Fig. 1h, i and Supplementary Fig. 1m). Together, these results demonstrate that the purine biosynthetic pathway is enhanced in MLL-AF9-driven LSCs.

### MYC induces enhanced purine metabolism in LSCs

Having identified the enhanced purine metabolism in LSCs, we sought to uncover the mechanism underlying the upregulated purine biosynthesis pathway. We analyzed the mRNA expression of key genes involved in purine synthesis in MLL-AF9-induced LSCs, bulk AML cells, as well as GMPs and normal WBM cells. The bulk AML cells isolated from MLL-AF9-induced AML mice exhibited higher expression of the key genes involved in purine biosynthesis, except *Impdh1*, when compared to GMPs and WBM cells (Fig. 2a). Notably, consistent with the elevated purine metabolite levels in LSCs, the majority of the purine biosynthetic genes, such as *Ppat, Pfas, Paics, Adss*, and *Gmps*, exhibited the highest expression in LSCs compared to bulk AML, GMPs and WBM cells (Fig. 2a). Additionally, we analyzed publicly available transcriptomics data from AML patients with a range of mutation profiles[23], where functional LSCs have been validated by xenotransplantation, and found that the majority of the purine biosynthetic genes were expressed at higher levels in LSCs compared to non-LSCs (Fig. 2b). These results indicate that increased purine biosynthetic gene expression is a conserved feature of LSCs.

To understand the mechanism by which LSCs increase the expression of purine biosynthetic genes, we first examined whether MLL-AF9 directly regulates the transcription of key components involved in purine biosynthesis. An analysis of a MLL-AF9 ChIP-seq dataset in MLL-AF9 expressing murine LSCs[24] revealed MLL-AF9 binding at the promoter of *Meis1*, a known target of MLL-AF9, but not at the promoters of the purine biosynthetic genes (Supplementary Fig. 2a), suggesting that the enhanced purine metabolism is unlikely due to the transcriptional regulation by MLL-AF9. We therefore postulated that transcription factors activated by MLL-AF9 might govern the expression of purine biosynthesis genes. Additionally, leveraging publicly available ChIP-seq datasets of murine hematopoietic cells from ChIP-Atlas (https://chip-atlas.org/), we identified 223 transcription factors (Supplementary Data 2) that bind to the promoter regions (-250 bp to +1000 bp of start codons) of genes involved in purine

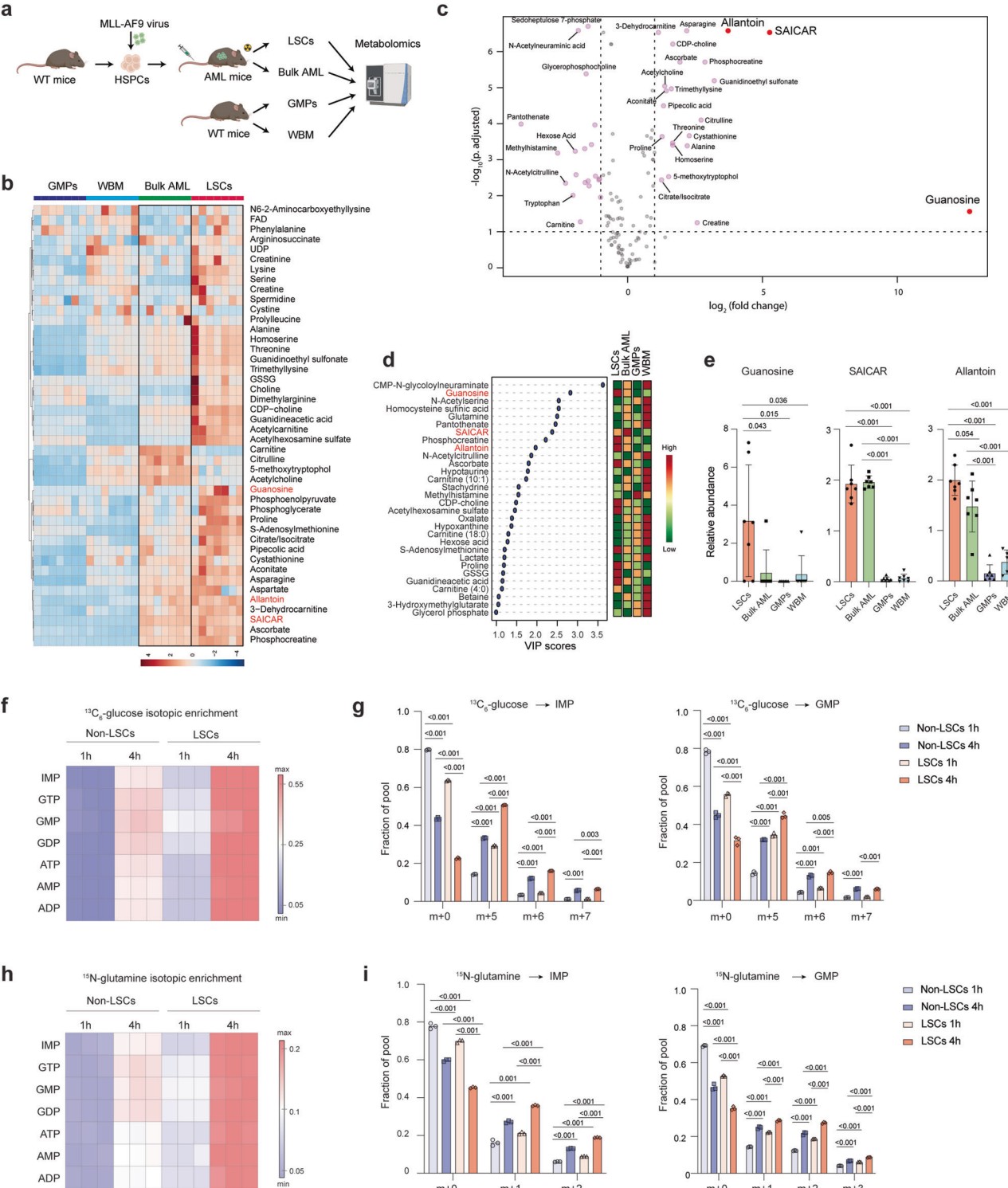

**Fig. 1 | Purine metabolism is enhanced in LSCs. a** Schematic flowchart of the targeted metabolomics analysis in LSCs and bulk AML from MLL-AF9-induced murine AML and wild-type (WT) GMPs and whole bone marrow (WBM) cells from C57BL/6 mice. Created in BioRender[67]. **b** Heatmap depicting the relative abundance of metabolites enriched in bulk AML and LSCs (see Supplementary Fig. 1a, Group 2) detected in LSCs, bulk AML, GMPs, and WBM cells (n = 7). **c** A volcano plot showing the abundance of metabolites in LSCs compared to GMPs. The metabolites in red are involved in purine biosynthesis and enriched in LSCs. **d** Variable importance in the projection (VIP, left panel) scores of metabolites detected in LSCs, bulk AML, GMP, and WBM cells. The color code on the right indicates the level of metabolites in each group (red: high, green: low). **e** Relative abundance of guanosine, SAICAR,

and allantoin in LSCs, bulk AML, GMPs, and WBM cells (n = 7, biologically independent samples). **f, h** Heatmap depicting the isotopic enrichment of purine biosynthesis intermediates in LSCs and non-LSCs treated with $^{13}C_6$-glucose (**f**) or amide-$^{15}$N-glutamine (**h**) for 1 and 4 h (n = 3). **g, i** Fractional labeling of IMP (left panel) and GMP (right panel) in LSCs and non-LSCs treated with $^{13}C_6$-glucose (**g**) or amide-$^{15}$N-glutamine (**i**) for 1 and 4 h (n = 3, biological samples). h, hour. All data are represented as mean ± standard deviation (SD). p values in this figure were calculated by unpaired, two-tailed Welch's t-test and Benjamini-Hochberg correction (**c**) and ANOVA with multiple comparisons analysis using Bonferroni correction post hoc analyses (**e, g,** and **i**). See also Supplementary Fig. 1 and Data 1. Source data are provided as a Source Data file.

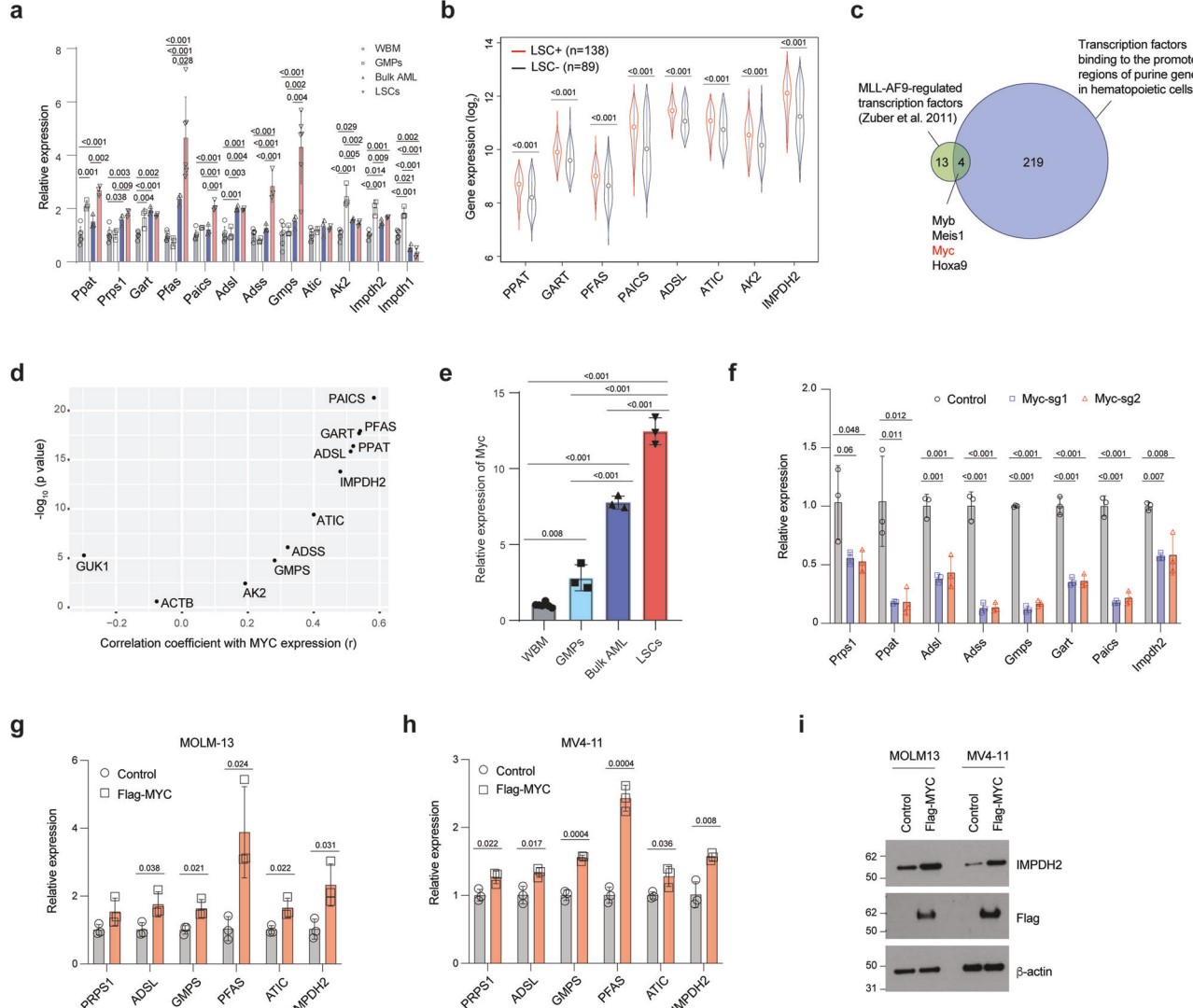

**Fig. 2 | MYC regulates the expression of purine biosynthetic genes in LSCs.**
**a** Relative expression of purine biosynthetic genes in murine WBM cells ($n = 6$), GMPs ($n = 3$), MLL-AF9-induced AML ($n = 3$), and LSCs ($n = 6$, biologically independent samples). **b** Relative expression of purine biosynthetic genes from human AML LSC+ ($n = 138$) and LSC− ($n = 89$) (GEO: GSE76009). The box plot presents an interquartile range, and the whiskers show a 95% confidence interval. **c** A schematic showing the overlap between 17 transcription factors regulated by MLL-AF9 and 223 transcription factors that bind to the promoters of genes involved in purine biosynthesis. **d** Correlation between the expression of *MYC* and genes involved in purine biosynthesis. r, correlation coefficient. **e** Relative expression of *Myc* in WBM cells ($n = 6$), GMPs ($n = 3$), MLL-AF9-induced AML ($n = 3$), and LSCs ($n = 3$, biologically independent samples). **f** Relative expression of purine biosynthetic genes in

Cas9-expressing LSCs expressing sgRNAs targeting control or *Myc* ($n = 3$, biologically independent samples). **g**–**i** Relative expression of purine biosynthetic genes in MOLM-13 (**g**), MV4-11 cells (**h**), and immunoblotting analysis of IMPDH2, Flag-MYC, and β-actin on separate membranes in MOLM-13 and MV4-11 cells expressing a control vector or Flag-MYC (**i**) ($n = 3$, biologically independent samples). The immunoblots are representative from at least two independent experiments. All data are represented as mean ± SD. p values in this figure were calculated by Pearson/Spearman correlation test (**d**), unpaired, two-tailed Student's t-test (**b**, **g**, and **h**) or ANOVA with multiple comparisons analysis using Bonferroni correction (**a** and **e**) or Dunnett's (**f**) post hoc analyses. See also Supplementary Fig. 2. Source data are provided as a Source Data file.

biosynthesis. Among the 223 transcription factors, four (Myb, Meis1, Myc, and Hoxa9) are among the 17 transcription factors whose expression is dependent on MLL-AF9 expression (Fig. 2c)[25]. We focused on Myc given its role in upregulating purine biosynthesis in solid tumors[21,26]. Analysis of the dataset that led to the discovery of 17-gene stemness score in AML[23] revealed a significant positive correlation between the expression of *MYC* and all except one (*GUK1*) of purine biosynthesis genes (correlation $r = 0.192 - 0.583$, $p < 0.05$) (Fig. 2d and Supplementary Fig. 2b). *Myc* expression was highest in MLL-AF9-induced murine LSCs compared to bulk AML and normal WBM cells (Fig. 2e). To examine the role of MYC in regulating the expression of purine biosynthetic genes, we deleted *Myc* or the control

loci *Rosa26* in MLL-AF9+ LSCs by CRISPR/Cas9. As expected, deletion of *Myc* resulted in reduced expression of purine biosynthetic genes (Fig. 2f). The reduced expression of purine biosynthetic genes was also found in MOLM-13 and THP-1 cells after deletion of *MYC* (Supplementary Fig. 2c, d). On the contrary, overexpression of FLAG-tagged-*MYC* in MOLM-13 and MV4-11 cells led to increased expression of the purine biosynthetic genes (Fig. 2g, h). Immunoblotting analysis confirmed the increased protein level of IMPDH2, a rate-limiting enzyme in the de novo biosynthesis of guanine nucleotides (Fig. 2i), in *MYC* overexpressing cells. Collectively, these results indicate that MYC serves as a functional mediator linking oncogenic MLL-AF9 with enhanced purine metabolism in LSCs.

## The purine biosynthetic pathway is a genetic dependency of AML

We previously performed a whole genome CRISPR dropout screen in MOLM-13 cells and identified regulators of NAD+ metabolism as AML vulnerabilities[27]. Upon re-analysis of this screening result, we found that purine metabolism ranked among the top of the GO terms for the dropout genes (Supplementary Fig. 3a). sgRNA against genes in the purine biosynthesis pathway, including *PPAT*, *GART*, *PFAS*, *ADSL*, *ATIC*, *GMPS*, and *IMPDH2*, exhibited a time-dependent depletion in the screen (Fig. 3a), indicating the essential role of purine metabolism in AML. Additionally, the cancer dependency map[28,29] showed that human AMLs are highly dependent on the purine biosynthetic genes compared to other cancer types (Fig. 3b). To validate the requirement of purine biosynthetic genes in human AML, we performed a competitive cell growth experiment using three AML cell lines, MOLM-13, THP-1, and HL-60. Deletion of these purine biosynthetic genes with two sgRNA sequences, with *MYC* included as a control, reduced the representation of sgRNA-expressing cells over a 17-day period (Fig. 3c and Supplementary Fig. 3b, c). Thus, the purine biosynthetic pathway is a genetic dependency of AML.

### Inhibition of purine biosynthetic pathway drives myeloid differentiation of LSCs

One of the key characteristics of AML is the block of myeloid differentiation. To investigate whether purine biosynthesis regulates myeloid differentiation of LSCs, we isolated LSCs from the MLL-AF9-induced AML mouse model and treated them with purine biosynthesis inhibitors mycophenolate mofetil (MMF), mycophenolic acid (MPA), or 6-mercaptopurine (6-MP). An immunosuppressant MMF is a prodrug of MPA and inhibits guanine nucleotide biosynthesis by targeting IMPDHs. 6-MP is a purinergic antimetabolite agent that is not only incorporated into nucleic acid chains but can also inhibit de novo purine biosynthesis. Treating LSCs with these three agents resulted in myeloid differentiation, as evidenced by flow cytometry showing increased expression of mature myeloid cell makers CD11b and Gr-1 (Fig. 3d and Supplementary Fig. 3d). Morphological examination by Giemsa stain revealed dose-dependent myeloid differentiation by purine biosynthesis inhibitors, characterized by an increased cytoplasm/nucleus ratio (Fig. 3e). We observed that the differentiated myeloid cells derived from LSCs treated with MMF appeared to have engulfed surrounding dead cells and/or debris (Fig. 3e). To establish the phagocytic activity of LSC-derived myeloid cells, we incubated MMF-exposed murine LSCs with a group B *Streptococcus* strain COH1 expressing GFP or stained with a fluorescent dye (Fig. 3f and Supplementary Fig. 3e, f). While approximately 20% of cultured LSCs displayed a phagocytic ability to engulf bacteria without MMF treatment, MMF treatment of LSCs resulted in a dose-dependent increase in the percentage of phagocytic myeloid cells, reaching more than 60% after 1 day and 80% after 2 days of MMF treatment (Fig. 3f and Supplementary Fig. 3e, f). These data demonstrate that suppression of the purine biosynthesis pathway leads to myeloid differentiation of LSCs into mature myeloid cells with phagocytic ability.

### Reduced guanine nucleotides block myeloid differentiation

We focused on the guanine nucleotide biosynthetic pathway given the potent effects that MMF and MPA had on AML differentiation. These drugs inhibit IMPDH and block the guanine nucleotide biosynthetic pathway. Consistently, we observed that MMF treatment decreased guanine nucleotide levels, such as guanosine mono and dinucleotides, in murine LSCs (Fig. 3g). Notably, the addition of guanosine, a downstream metabolite of IMPDH (Supplementary Fig. 1j), to MMF-treated LSCs reversed the reduction of guanosine mono and dinucleotides, suggesting that guanosine was converted to guanosine mononucleotides through the purine salvage pathway in LSCs (Fig. 3g). While guanosine supplementation in untreated LSCs did not affect the

expression of myeloid differentiation markers CD11b and Gr-1, guanosine treatment of MMF-treated LSCs remarkably reversed myeloid differentiation (Fig. 3h), establishing that reduced guanine nucleotides promote myeloid differentiation of AML. LSCs treated with MMF also exhibited decreased cell cycling and increased apoptosis, both of which were rescued by guanosine (Fig. 3i, j and Supplementary Fig. 3h–i). To demonstrate the specificity of IMPDH inhibition in causing myeloid differentiation of LSCs, we treated LSCs with two additional IMPDH inhibitors, mizoribine and tiazofurin. Similar to MMF, treatment of LSCs with these two agents increased myeloid differentiation, decreased cell cycle, and increased apoptosis, all of which were rescued by guanosine (Supplementary Fig. 3j–n). These results indicate that guanine nucleotide biosynthesis regulates myeloid differentiation, cell cycle, and apoptosis of LSCs.

We then took a genetic approach to inhibit the guanine biosynthetic pathway by deleting IMPDH1/2 in LSCs. To achieve this, we introduced lentiviral sgRNAs targeting either *Impdh1*, *Impdh2*, or *Rosa26* into LSCs isolated from MLL-AF9-driven AML generated from Cas9-inducible mice[30]. Treatment with doxycycline to induce Cas9 reduced the fraction of LSCs expressing sgRNA targeting *Impdh2* but not *Impdh1* or *Rosa26* (Fig. 3k). Immunoblotting confirmed the reduction of IMPDH1 and IMPDH2 protein levels (Supplementary Fig. 3o). Deletion of *Impdh2* in LSCs induced myeloid differentiation, reduced cells in S-phase, and increased apoptosis (Fig. 3l and Supplementary Fig. 3p, q). These data establish that genetic and pharmacological suppression of the guanine biosynthetic pathway drives myeloid differentiation of LSCs.

### *MLL*r AMLs are sensitive to guanine biosynthesis inhibition

To determine whether AMLs with particular genetic lesions are more dependent on the purine biosynthesis pathway, we treated a genetically diverse panel of human AML cell lines with MMF, MPA, and 6-MP (Supplementary Fig. 3r, s). Our analysis unveiled that AML cells carrying MLL fusions, such as MOLM-13 (MLL-AF9), THP-1 (MLL-AF9), MV4-11 (MLL-AF4) and NOMO1 (MLL-AF9), are relatively more sensitive to the inhibitors compared to AML cells without MLL fusions (Supplementary Fig. 3r, s). Consistently, expression of MLL-ENL fusion rendered LSK cells more sensitive to MMF compared to MOZ-TIF2 fusion as assessed by myeloid differentiation (Supplementary Fig. 3t). In addition, MMF treatment induced myeloid differentiation, as indicated by the increased expression of CD11b, CD14, CD33, and CD66b, predominantly in AML cell lines with *MLL* fusions (Fig. 3m and Supplementary Fig. 3u, v). We also treated human AML cells isolated from patient-derived xenograft (PDX) models and found that *MLL*r AML cells were more sensitive to MMF treatment than non-*MLL*r AML (Fig. 3n, Supplementary Fig. 3w and Data 3). Collectively, these results indicate that while AML cells display varied responses to guanine biosynthesis inhibition based on their genetic alterations, *MLL*r AML are among the most sensitive subtypes.

We next investigated whether inhibiting guanine biosynthesis affects LSC function. We transiently treated LSCs with or without MMF for 12 h and plated them on methylcellulose media without MMF (Fig. 4a). Approximately 25% of LSCs formed packed colonies, characteristic of MLL-AF9-induced AML[31], and this efficiency remained consistent across additional two rounds of plating (Fig. 4b, c). MMF treatment significantly reduced both the size and cell number within LSC-derived colonies (Fig. 4c). The resulting AML colonies that formed after MMF treatment had fewer cells and these cells exhibited myeloid differentiation (Fig. 4d–f). We also transplanted varying numbers of LSCs after ex vivo MMF treatment (Fig. 4a). This pre-treatment with MMF resulted in prolonged leukemia-free survival and a reduction in AML burden in the peripheral blood of recipient mice (Fig. 4g, h and Supplementary Fig. 4a).

We then sought to examine the role of guanine biosynthesis in the initiation and maintenance of AML. To this end, cohorts of MLL-AF9

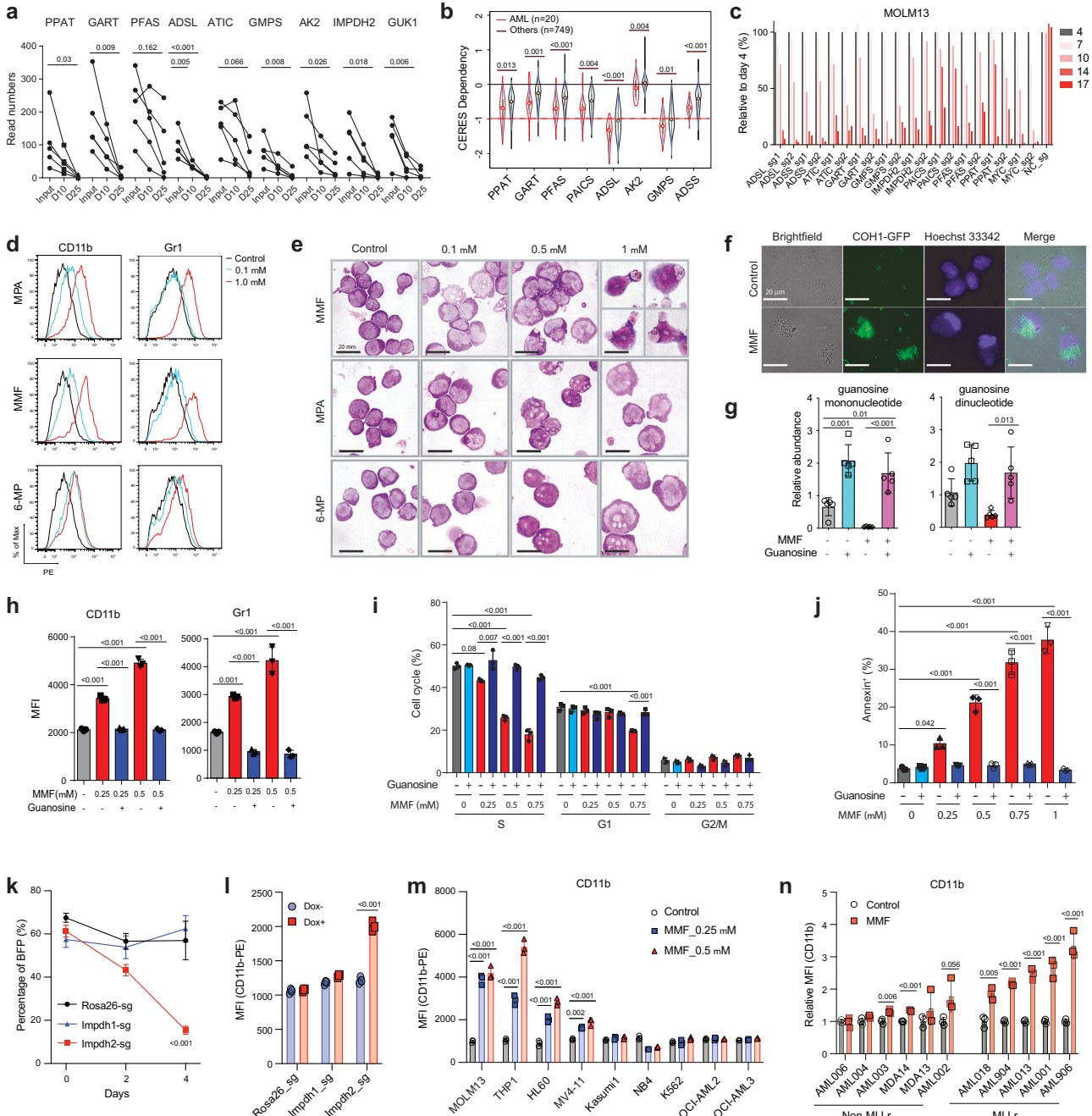

**Fig. 3 | Inhibition of purine biosynthesis promotes myeloid differentiation of LSCs in vitro. a** The read numbers of sgRNAs against purine biosynthetic genes at day 25 compared to day 10 and input from a whole-genome CRISPR screen in MOLM-13 cells; ($n = 5$, individual sgRNA). Each dot represents one sgRNA. D, day. **b** Cancer cell line dependency scores (CERES) of purine biosynthetic genes in AML ($n = 20$) and other cancers ($n = 749$) from DepMap. The box plot presents an interquartile range, and the whiskers show a 95% confidence interval. **c** Competitive growth assays with Cas9-expressing MOLM-13 cells that express sgRNAs against negative control (NC, negative control), positive control (*MYC*), and genes involved in the purine biosynthetic pathway over 17 days. The percentages of sgRNA-expressing cells were normalized to those on day 4 after transduction. **d** Flow cytometry histograms of mature myeloid cell markers CD11b and Gr-1 expression in murine LSCs upon treatment with MPA, MMF, or 6-MP (0.1-1 μM) for 24 h. MPA, mycophenolic acid; 6-MP, 6-mercaptopurine. **e** Wright–Giemsa staining of LSCs showing myeloid differentiation upon treatment with MMF, MPA, and 6-MP (0.1-1 μM) for 24 h. Scale bar, 20 μm. **f** Fluorescence images showing engulfment of GFP-labeled Streptococcus agalactiae COH1 by LSC-derived myeloid cells after exposure to MMF (1 μM) for 24 h. Scale bar, 20 μm. **g** Relative abundance of guanosine mono

and dinucleotides in LSCs treated with MMF (0.25 μM) or guanosine (100 μM) alone or in combination for 2 h ($n = 5$, biologically independent samples). **h** Mean fluorescence intensity (MFI) of mature myeloid cell marker CD11b (left panel) and Gr-1 (right panel) in LSCs treated with MMF (0.25-0.5 μM) alone or in combination with guanosine (100 μM) for 24 h ($n = 3$, biologically independent samples). **i, j** Cell cycle (**i**) and apoptosis (**j**) analyses of LSCs upon treatment with MMF (0.25-1 μM) alone or in combination with guanosine (100 μM) for 24 h ($n = 3$, biologically independent samples). **k, l** Percentage of BFP (present in sgRNA vector) (**k**) and MFI of CD11b (**l**) in Cas9-expressing LSCs after the transduction of sgRNA targeting *Rosa26, Impdh1*, or *Impdh2* genes for 4 days ($n = 3$, biologically independent samples). **m** MFI of CD11b in a panel of AML cell lines treated with MMF (0.25-0.5 μM) for 3 days ($n = 3$, biologically independent samples). **n** Relative MFI of CD11b in PDX samples treated with MMF (1 μM) for 6 days ($n = 3$, biological samples). All data are represented as mean ± SD. p values in this figure were calculated by unpaired, two-tailed Student's t-test (**b**, **l** and **n**) or ANOVA with multiple comparisons analysis using Bonferroni correction post hoc analyses (**a**, **g**–**k** and **m**). See also Supplementary Fig. 3. Source data are provided as a Source Data file.

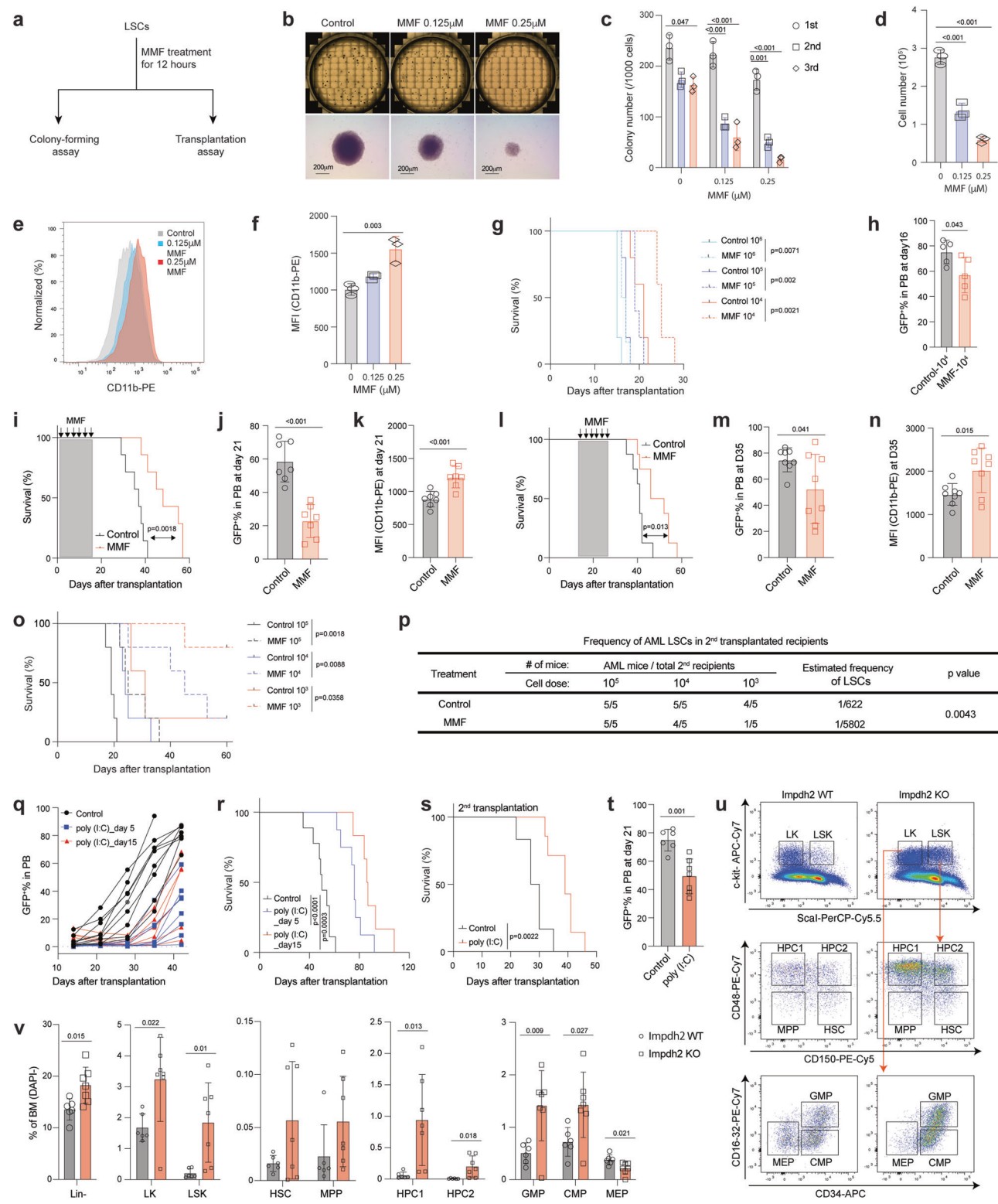

Frequency of AML LSCs in 2nd transplantated recipients

| Treatment | AML mice / total 2nd recipients | | | Estimated frequency of LSCs | p value |
|---|---|---|---|---|---|
| Cell dose: | 10⁵ | 10⁴ | 10³ | | |
| Control | 5/5 | 5/5 | 4/5 | 1/622 | |
| MMF | 5/5 | 4/5 | 1/5 | 1/5802 | 0.0043 |

AML recipient mice were treated with MMF either from days 1 to 14 ("early") or from days 14 to 28 ("delayed") after transplantation (Supplementary Fig. 4b). Both MMF treatment regimens significantly extended the survival of AML recipient mice, reduced the frequency of AML cells, and induced myeloid differentiation in the peripheral blood (Fig. 4i−n). The body weight of the mice was not significantly affected by MMF treatment in both conditions (Supplementary Fig. 4c, d). To assess the frequency of LSCs, we performed a limiting dilution assay (LDA) by transplanting limiting doses of AML cells isolated from AML

recipient mice that had been subjected to the MMF "early" treatment. This revealed that MMF treatment reduced the frequency of LSCs by 9-fold (1 in 622 cells in control versus 1 in 5,802 cells in MMF) (Fig. 4o, p and Supplementary Fig. 4e). These results establish that the inhibition of guanine nucleotide biosynthesis suppresses AML initiation and progression by impairing LSC function.

MMF is used in the clinic as an immunosuppressant, but its effects on hematopoiesis are not well-defined. To address this question, we treated wild-type mice with 100 mg/kg of MMF for two weeks and

**Fig. 4 | Purine biosynthesis is required for LSC maintenance in vivo. a** Schematic flowchart depicting colony-forming and transplantation assays in LSCs treated with or without MMF. **b** Representative images of colonies from 6-well plates and individual colonies derived from control and MMF (0.125-0.25 μM)-treated LSCs. Scale bar, 200μm. **c** Number of colonies derived from LSCs after exposure to MMF treatment (0.125-0.25 μM) during serial plating (*n* = 3). **d–f** Cell number (**d**), flow cytometry histograms (**e**), and MFI of myeloid differentiation marker CD11b (**f**) from the LSC-derived colonies treated with or without MMF (0.125-0.25 μM) after the first plating (*n* = 3, biologically independent samples). **g** Kaplan-Meier analysis of survival of recipient mice transplanted with different numbers of LSCs exposed to MMF (0.125 μM) for 12 h ex vivo (*n* = 5). **h** Frequency of GFP⁺ AML cells in the peripheral blood (PB) of recipient mice receiving 10⁴ LSCs after exposure to MMF (0.125 μM) at 16 days post-transplantation (*n* = 5). **i, l** Kaplan-Meier analysis of survival of mice transplanted with MLL-AF9-induced AML and treated with either vehicle or MMF (100 mg/kg/day) at days 1 to 14 post-transplantation (**i**, *n* = 7) or days 15 to 28 post-transplantation (**l**, *n* = 8). **j, k, m, n** Frequency of GFP⁺ AML cells

(**j, m**) and MFI of CD11b (**k, n**) in the PB of AML recipient mice treated with either vehicle or MMF as in (**i**) (**j, k**, *n* = 7) or as in (**l**) (**m, n**, *n* = 8). **o, p** Kaplan-Meier analysis of survival (**o**) and estimated frequency of LSCs (**p**) of MLL-AF9-induced AML mice that were treated with MMF as in (**i**) (*n* = 5). **q, r** Frequency of GFP⁺ AML cells in the PB (**q**) and survival analysis (**r**) of recipient mice transplanted with MLL-AF9⁺ *Mx1-Cre; Impdh2^{fl/fl}* AML cells and treated with poly(I:C) at days 5 or 15 after transplantation. **s, t** Kaplan-Meier analysis of survival (**s**) and frequency of GFP⁺ AML cells in PB (**t**) of secondary recipient mice transplanted with AML cells from moribund mice in (**r**). **u, v** Representative flow cytometry dot plots (**u**) and frequencies of Lin⁻ and HSPCs (**v**) in the bone marrow of *Impdh2* WT (*n* = 6) and KO (*n* = 7) mice. All data are represented as mean ± SD. p values in this figure were calculated by unpaired, two-tailed Student's t-test (**h, j, k, m, n, t** and **v**), ANOVA with multiple comparisons analysis using Dunnett's post hoc analyses (**c, d**, and **f**), or log-rank test (**g, i, l, o, r** and **s**). See also Supplementary Fig. 4. Source data are provided as a Source Data file.

examined hematopoietic cells in the bone marrow, spleen, thymus, and peripheral blood (Supplementary Fig. 4f). MMF treatment reduced spleen cellularity and weight, without obvious changes in the body weight (Supplementary Fig. 4g–i). Analysis of the bone marrow revealed no changes in the frequency of HSPCs except for a reduction in CD150⁻CD48⁻LSK MPPs (Supplementary Fig. 4j). MMF treatment did not affect myeloid, B-, and T-cells in the blood but increased red blood cells and hemoglobin (Supplementary Fig. 4k, l). Additionally, mice receiving MMF treatment exhibited reduced CD4⁺ and CD8⁺ T-cells in the thymus, without changes in T-cell progenitors (Supplementary Fig. 4m). These findings demonstrate that disrupting the guanine biosynthetic pathway has minimal impact on normal hematopoiesis.

To further examine the dependency of AML on the guanine biosynthesis pathway, we crossed *Impdh2^{fl}* mice with *Mx1-Cre* mice to generate *Mx1-Cre; Impdh2^{fl/fl}* mice. HSPCs isolated from these mice were retrovirally transduced with MLL-AF9 and transplanted. Upon the development of primary AML, we transplanted 5,000 AML cells into recipient mice and treated them with either saline or poly-inosinic:polycytidylic acid (poly(I:C)) to induce Cre (hereafter referred to as *Impdh2* KO). We injected poly(I:C) either from day 5 or day 15 after transplantation and confirmed efficient deletion of *Impdh2* in peripheral blood AML cells (Supplementary Fig. 4n, o). Deletion of *Impdh2* from engrafted AML significantly reduced the frequency of AML cells in the peripheral blood and delayed leukemogenesis (Fig. 4q, r and Supplementary Fig. 4p). *Impdh2* KO AML exhibited delayed leukemogenesis and reduced AML burden in the peripheral blood upon serial transplantation (Fig. 4s, t and Supplementary Fig. 4n). The residual AML cells after poly (I:C) injection retained the *Impdh2* flox allele, indicating that cells that escaped *Impdh2* deletion outcompeted *Impdh2* KO cells (Supplementary Fig. 4o). These results establish that *Impdh2* is critical for LSC activity.

We also investigated the impact of *Impdh2* deletion on normal hematopoiesis with our inducible *Impdh2* KO mice model (Supplementary Fig. 4q). While the cellularity of bone marrow remained unaffected 3-4 months post poly(I:C) injection (Supplementary Fig. 4r), we observed an increased frequency of lineage⁻ (Lin⁻), Lin⁻c-kit⁺(LK), Lin⁻c-kit⁺Sca-1⁺(LSK), hematopoietic progenitors 1/2 (HPC1/2), myeloid progenitors such as GMPs and common myeloid progenitors (CMPs), along with a decreased frequency of megakaryocyte-erythroid progenitors (MEPs) (Fig. 4u, v). The frequency of B220⁺ B-cells in both peripheral blood and bone marrow was reduced, while the frequency of Mac1⁺Gr1⁺ myeloid cell was increased in the bone marrow after *Impdh2* deletion (Supplementary Fig. 4s). Furthermore, *Impdh2* deletion resulted in decreased numbers of white blood cells, red blood cells, hemoglobin and platelets (Supplementary Fig. 4t). Overall, these findings suggest that *Impdh2* deletion promotes the expansion of myeloid progenitors and mature cells while restricting erythroid and lymphoid lineage development.

To further examine the role of *Impdh2* in HSC function, we performed competitive transplantation assays in which WBM cells from CD45.2 *Mx1-Cre; Impdh2^{fl/fl}*, *Mx1-Cre; Impdh2^{fl/+}* or *Impdh2^{fl/fl}* mice (hereafter referred to as KO, Het, and WT, respectively) were mixed 1:1 with WBM cells from CD45.1 and transplanted into CD45.1 recipients (Supplementary Fig. 4u). We then injected the recipient mice with poly (I:C) to delete *Impdh2* five days after transplantation and analyzed the hematopoietic contribution by CD45.2 donor-derived cells. While *Impdh2* heterozygosity did not affect hematopoietic reconstitution, homozygous deletion of *Impdh2* reduced multi-lineage reconstitution (Supplementary Fig. 4v). *Impdh2* KO cells contributed significantly less to the hematopoietic cells residing in different organs, including the liver, spleen, thymus, and bone marrow HSPCs, than Het or WT cells (Supplementary Fig. 4w, x). Thus, while *Impdh2* is largely dispensable for steady-state hematopoiesis, it is essential for hematopoietic regeneration after transplantation. Collectively, these data establish that *Impdh2* is indispensable for both normal HSC function and malignant LSC activity, but that a therapeutic window exists to target AML by pharmacological inhibition of the guanine biosynthesis pathway.

**Inhibition of guanine biosynthesis reduces rRNA synthesis**

We then sought to understand the mechanisms by which inhibition of guanine biosynthesis promotes LSC differentiation. IMPDH inhibition reduces guanosine nucleotides (Fig. 3g) used for DNA and RNA synthesis. Flux analysis with ¹³C₆-glucose in murine LSCs revealed higher labeling in GTP than dGTP, suggesting a higher GTP synthesis rate (Fig. 5a), consistent with the higher utilization rate of GTP compared to other nucleotides in rRNA synthesis[32]. Thus, we reasoned that the myeloid differentiation of LSCs caused by inhibiting guanine nucleotide biosynthesis could be associated with disruption of rRNA transcription. To test this, we examined the impact of guanine biosynthesis inhibition on rRNA transcription. Compared to *Gapdh* pre-mRNA, the level of pre-rRNA was significantly reduced by MMF treatment, which was rescued by guanosine supplementation (Fig. 5b). Additionally, disruption of guanine biosynthesis through *IMPDH2* deletion in MOLM-13 cells resulted in reduced rRNA synthesis, which was reversed by the addition of guanosine (Supplementary Fig. 5a). These data establish the critical role of guanine biosynthesis in RNA pol I-mediated rRNA transcription. If rRNA biosynthesis is a major consumer of guanine nucleosides, then inhibition of RNA polymerase I that is responsible for rRNA transcription should alleviate the reduction of guanine nucleosides caused by MMF. We thus exposed LSCs to MMF and/or CX-5461, an inhibitor of RNA polymerase I, and measured guanine nucleosides levels (Fig. 5c). Compared to lactate, an internal control unaffected by MMF or CX-5461, MMF significantly reduced guanosine mono, di, and trinucleotides levels while increasing AMP/ADP levels. Inhibition of rRNA synthesis by CX-5461 blocked the

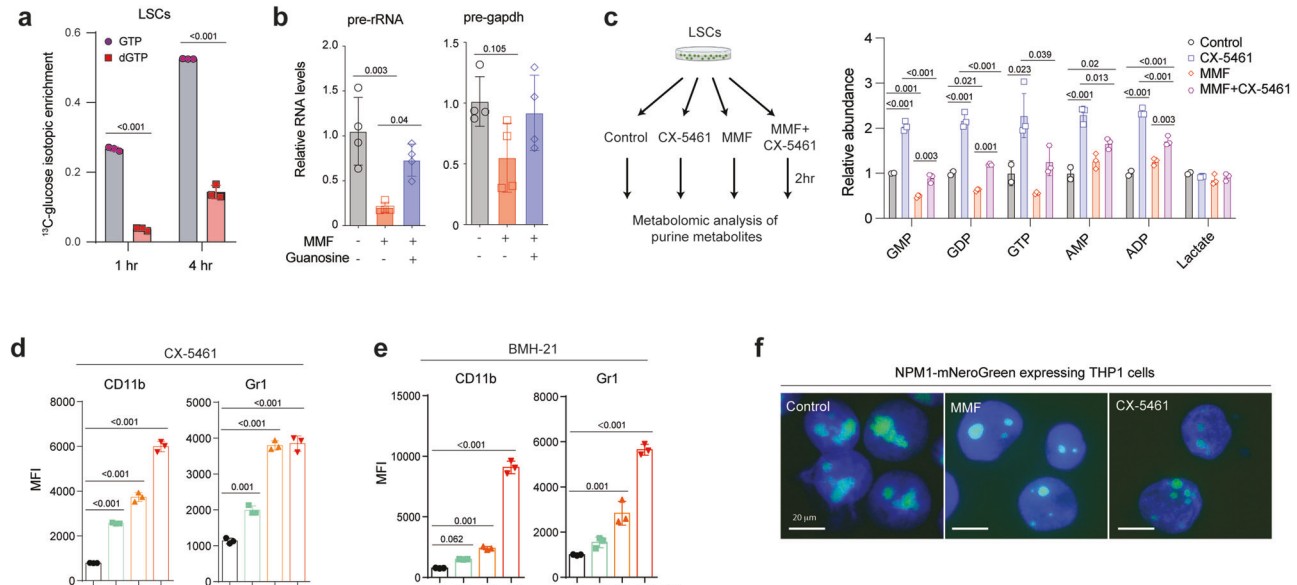

**Fig. 5 | Inhibition of rRNA transcription drives myeloid differentiation of LSCs.** **a** Isotopic enrichment of GTP and dGTP in murine LSCs treated with $^{13}C_6$-glucose for 1 and 4 h ($n = 3$, biological samples). **b** Relative expression of pre-rRNA and pre-*Gapdh* in LSCs treated with vehicle, MMF (0.25 µM) or guanosine (100 µM) for 4 h ($n = 3$, biologically independent samples). **c** Schematic flowchart created in Biorender[68] and relative abundance of guanosine mono, di, and trinucleotides, AMP/ADP, and lactate (internal control) of LSCs treated with control, CX-5461 (0.1 µM), MMF (0.25 µM), or CX-5461 (0.1 µM) plus MMF (0.25 µM) for 2 h (right panel) ($n = 2$–3). **d**, **e** MFI of myeloid differentiation marker CD11b (left panel) and Gr-1 (right panel) in LSCs treated with CX-5461 (**d**, 25–250 nM) or BMH-21 (**e**, 50-500 nM) for 24 h ($n = 3$, biologically independent samples). **f** Fluorescence images of THP-1 cells expressing mNeonGreen from the *NPM1* locus (NPM1-mNeonGreen) treated with control vehicle, MMF (0.25 µM) or CX-5461 (0.1 µM) for 24 h. Scale bar, 20 µm. All data are represented as mean ± SD. p values in this figure were calculated by unpaired, two-tailed Student's t-test (**a**) or ANOVA with multiple comparisons analysis using Bonferroni correction (**b** and **c**) or Dunnett's (**d** and **e**) post hoc analyses. See also Supplementary Fig. 5. Source data are provided as a Source Data file.

utilization of guanine nucleosides and normalized the reduced guanine nucleosides levels caused by MMF (Fig. 5c). Furthermore, inhibition of rRNA transcription by CX-5461 or BMH-21, another RNA polymerase I inhibitor that blocks transcriptional initiation and elongation, induced myeloid differentiation of LSCs (Fig. 5d, e and Supplementary Fig. 5b), phenocopying the effects of inhibiting guanine nucleotide biosynthesis. We then tested whether dysregulated rRNA transcription and processing accompanies morphological changes of the nucleoli. To this end, we knocked in a monomeric NeonGreen (mNeonGreen) cassette in the *NPM1* locus of THP-1 cells. Both MMF and CX-5461 treatment reduced the size of NPM1-labeled nucleoli (Fig. 5f). These data demonstrate that inhibition of nucleolar rRNA synthesis, either directly by inhibiting RNA polymerase I or by inhibiting guanine biosynthesis, promotes myeloid differentiation of LSCs.

### Inhibition of guanine biosynthesis reduces MLL-AF9 LSC gene expression

To examine how inhibition of guanine metabolism impacts the transcriptome of *MLL*r AML, we performed RNA sequencing (RNA-seq) with or without a spike-in control (Supplementary Fig. 6a). We isolated LSCs directly from MLL-AF9-induced AML mice and then immediately treated them with MMF for 16 h, either with or without guanosine supplementation. Analysis of gene expression changes without the spike-in control yielded only 576 and 458 genes that were increased or decreased by MMF treatment, respectively. In contrast, incorporating a spike-in control revealed a greater number of differentially expressed genes (DEG) upon MMF treatment compared to control (Supplementary Fig. 6b). Specifically, MMF downregulated 9,059 genes while it upregulated only 39 genes in LSCs (adjusted $p < 0.05$, $\log_2$(fold change)>1 or <-1) (Fig. 6a, b and Supplementary Data 4–5). Gene set enrichment analysis (GSEA) analysis revealed downregulation of MLL-AF9 target genes[25] and LSCs signature[33] (Fig. 6c and S6c). Although

only a subset of MMF-downregulated genes (1,049 out of 9,059) was rescued by guanosine (Fig. 6b, Supplementary Data 4 and 6), guanosine reversed the MMF-mediated downregulation of MLL-AF9 target[25] and LSCs gene[33] sets (Fig. 6d, e). Quantitative PCR (qPCR) confirmed the downregulation of MLL-AF9 target genes, including *Hoxa9*, *Meis1*, *Myc*, and *Cdk6*, as well as the upregulation of *Itgam* by MMF (Supplementary Fig. 6d). The reduced *Hoxa9* and *Meis1* expression by MMF were rescued by guanosine supplementation (Supplementary Fig. 6e). Gene ontology (GO) analysis unveiled downregulation of cell cycle, mRNA processing, rRNA processing, ribosome biogenesis, translation initiation, and chromatin organization/remodeling by MMF (Fig. 6f). Upregulated GO terms induced by MMF were associated with immune system response, defense response to bacteria, and neutrophil chemotaxis/aggregation (Supplementary Fig. 6f). Most of these GO term changes were normalized by guanosine supplementation (Supplementary Fig. 6g, h). Guanine nucleotide biosynthesis thus plays a pivotal role in maintaining LSC gene expression programs.

### Inhibition of rRNA synthesis reduces MLL-AF9 target gene expression

Given the similar differentiation phenotypes caused by the inhibition of guanine nucleotide biosynthesis or rRNA transcription, we examined whether rRNA transcription affects the MLL-fusion dependent gene expression program similarly to purine biosynthesis inhibition. RNA-seq of MLL-AF9+ murine LSCs exposed to CX-5461 for 16 h ex vivo revealed 204 and 52 genes downregulated and upregulated, respectively (Supplementary Data 4 and 7). GSEA and qPCR analyses revealed that CX-5461 downregulates MLL-AF9 target genes[24] (Fig. 6g and Supplementary Fig. 6d). Intriguingly, we found downregulation of GO terms related to nucleosome assembly and transcription from RNA polymerase II promoter (Supplementary Fig. 6i), which could be the indirect effect of rRNA transcription inhibition. These data support the

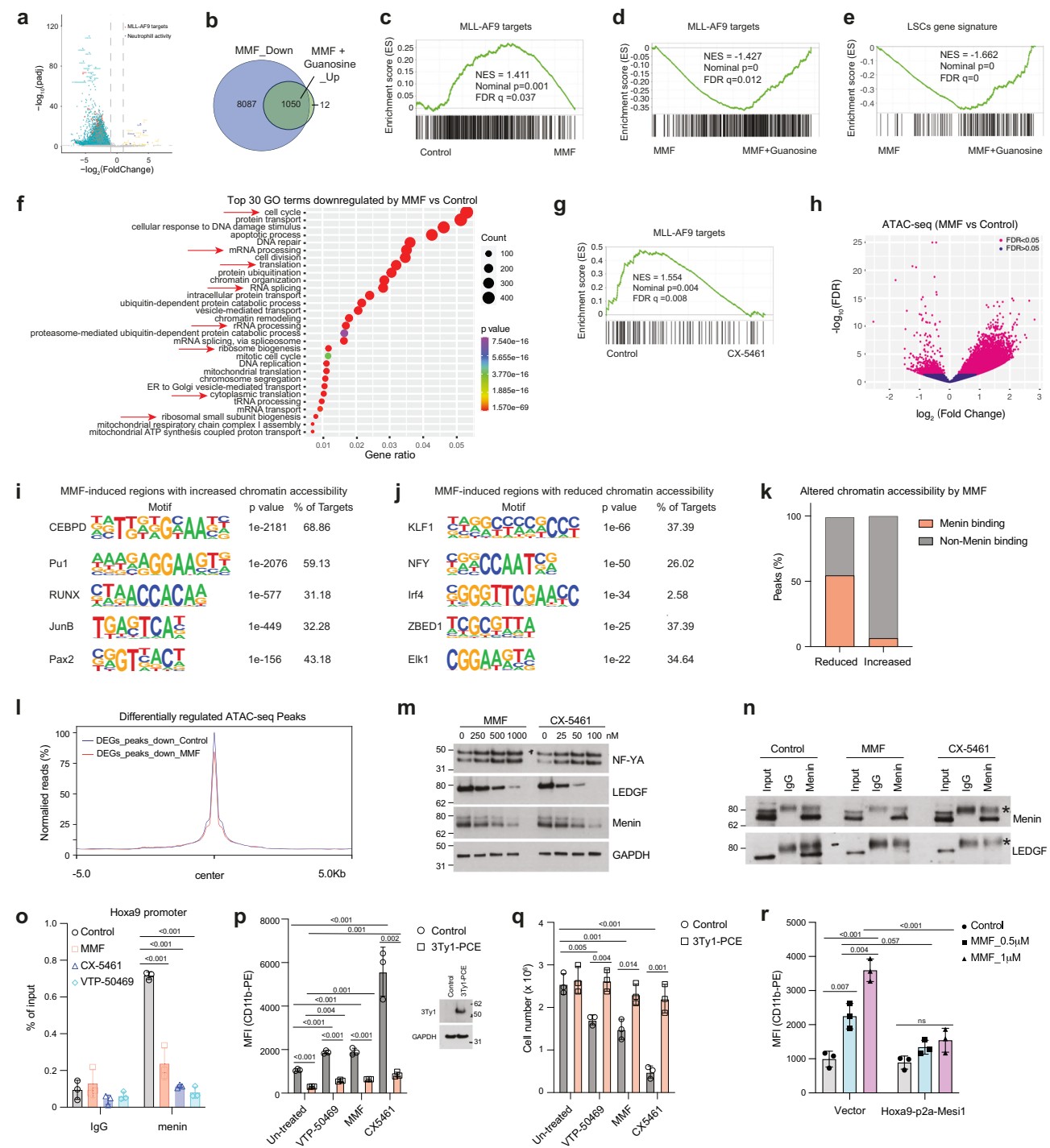

model that rRNA synthesis governed by purine biosynthesis is required to sustain the expression of MLL-fusion target genes.

## Inhibition of guanine nucleotide biosynthesis alters chromatin accessibility of LSCs

We then assessed whether inhibiting guanine nucleotide biosynthesis affects chromatin accessibility of LSCs by performing assay for transposase accessible chromatin sequencing (ATAC-seq) with murine LSCs treated with or without MMF (Supplementary Fig. 6j). MMF treatment resulted in increased accessibility of more sites compared to reducing accessibility, as evidenced by 13,398 regions becoming accessible and 1,127 regions becoming inaccessible by MMF treatment (Fig. 6h). Motif analysis of the chromatin regions that gained accessibility after MMF

treatment were enriched for C/EBPδ and PU.1 motif, while sites that lost accessibility were most enriched in KLF1 and NF-Y motifs (Fig. 6i, j). A recent report demonstrated that menin-occupied sites are significantly enriched in NF-Y motif, and that menin and NF-Y colocalize on chromatin[34]. This is consistent with the model that MMF treatment attenuates the chromatin recruitment of the menin-MLL complex. To determine whether loss of accessible regions containing NF-Y motifs leads to eviction of menin, we cross-referenced our ATAC-seq data with a published genome-wide menin occupancy profile[34]. Consistent with the possibility that the loss of occupancy at NF-Y motifs abrogates menin/MLL binding to chromatin, more than half (614 out of 1,127, 54.5%) of the sites that lost accessibility by MMF were menin binding sites in MLL-AF9-induced AML cells, while only 6.5% (875 out of 13,398)

**Fig. 6 | Inhibition of purine biosynthesis reduces LSC gene expression.**
**a** Volcano plot showing the genes downregulated, upregulated, and not significantly affected by MMF treatment in LSCs. Differentially Expressed Genes (DEGs) were identified using adjusted p < 0.05, and $\log_2$(fold change) >1 or <−1.
**b** Venn diagrams showing the number of genes downregulated by MMF compared to control, and genes upregulated by MMF plus guanosine compared to MMF treatment alone. **c** Gene set enrichment analysis (GSEA) plot showing negative enrichment of MLL-AF9 target genes[25] in MMF-treated versus control LSCs.
**d, e** GSEA plots showing positive enrichment of MLL-AF9 target genes[25] (**d**) and LSC gene signature[33] (**e**) in LSCs treated with MMF and guanosine versus those treated only with MMF. **f** Top 30 Gene Ontology (GO) terms for genes downregulated by MMF compared to control, as determined by DAVID functional annotation analysis.
**g** GSEA plot showing negative enrichment of genes directly regulated by MLL-AF9[24] in CX-5461-treated versus control LSCs. **h** Volcano plot showing altered chromatin accessibility of LSCs induced by MMF (0.25 μM) treatment for 16 h. **i, j** Motif analysis of MMF-induced regions with increased (**i**) and reduced (**j**) chromatin accessibility in LSCs by ATAC-seq. **k** Peaks of the altered chromatin accessibility in LSCs induced by MMF, overlapped with or without menin binding. **l** Meta profiles of ATAC-seq data from control and MMF-treated LSCs near genes downregulated by MMF. **m** Immunoblots of NF-YA, menin, LEDGF, and GAPDH on separate membranes in LSCs treated with control vehicle, MMF (250-1000 nM) or CX-5461 (25-

100 nM) for 16 h. A representative blot from at least two independent experiments is shown. **n** Immunoblots of menin and LEDGF in LSCs treated with control, MMF (0.5 μM), or CX-5461 (100 nM) for 16 h subjected to immunoprecipitation using either IgG or anti-menin antibodies. *, a non-specific band. A representative blot from at least two independent experiments is shown. **o** ChIP-qPCR analysis of menin occupation at the Hoxa9 promoter in LSCs treated with control, MMF (500 nM), CX-5461 (100 nM), or VTP-50469 (250 nM) (n = 3, biologically independent samples). **p** MFI of myeloid differentiation marker CD11b in MOLM-13 cells expressing vector control or 3Ty1-PCE, with or without treatment of VTP-50469 (125 nM), MMF (250 nM), and CX-5461 (100 nM) for 4 days (left panel). Immunoblots on the right show the protein level of 3Ty1-PCE and GAPDH on separate membranes (n = 3, biologically independent samples). **q** The effects of VTP-50469 (125 nM), MMF (250 nM), and CX-5461 (100 nM) on MOLM-13 cells expressing 3Ty1-PCE after 4 days (n = 3, biologically independent samples). **r** MFI of myeloid differentiation marker CD11b in THP-1 cells expressing vector control or Hoxa9-p2a-Meis1, with or without treatment of MMF (0.5-1 μM) for 4 days (n = 3, biologically independent samples). All data are represented as mean ± SD. p values in this figure were calculated by permutation test (**c–e** and **g**), hypergeometric test (**f, i** and **j**), unpaired, two-tailed student t-test (**p** and **q**) and ANOVA with multiple comparisons analysis using Dunnett's (**o**) or Bonferroni correction (**p, q** and **r**) post hoc analyses. See also Supplementary Fig. 6. Source data are provided as a Source Data file.

of the sites that increased accessibility by MMF are bound by menin (Fig. 6k). These results establish that inhibition of the guanine nucleotide biosynthetic pathway disrupts menin/MLL binding to chromatin while allowing myeloid transcription factors C/EBPδ and PU.1 to occupy the chromatin.

Transcriptional changes by MMF correlated with the changes in chromatin accessibility. Genes that were transcriptionally downregulated by MMF had reduced chromatin accessibility upon MMF treatment (Fig. 6l). Known menin/MLL-AF9 target genes that were downregulated by MMF, such as *Meis1*, *Myc*, and *Pbx3* had reduced chromatin accessibility in their promoter regions after MMF treatment (Supplementary Fig. 6k). In contrast, increased expression of *Itgam* by MMF was associated with increased accessibility of the promoter (Supplementary Fig. 6k). These data demonstrate that the transcriptional changes caused by MMF accompany altered chromatin accessibility.

### Inhibition of purine metabolism reduces the expression of LEDGF and menin
MLL and MLL-fusion oncoproteins form complexes with menin and LEDGF. In this complex, menin serves as an adaptor protein that links LEDGF to MLL[6]. Given that the chromatin occupancy of menin/NF-Y is reduced by MMF treatment, we examined the expression of NF-Y and the LEDGF/menin/MLL complex. MMF but not CX-5461 treatment reduced the transcript levels of *Nfya*, *Men1*, *Psip1* (encoding NF-YA, menin, and LEDGF, respectively) and other components of the MLL complex (Supplementary Fig. 6l). While the protein level of NF-YA remained unchanged, the levels of LEDGF and menin were significantly reduced by MMF or CX-5461 treatment in a dose-dependent manner (Fig. 6m). Cycloheximide (CHX) chase assay with or without CX-5461 or MMF did not show any difference in the half-life of LEDGF and menin protein (Supplementary Fig. 6m). While protein translation was globally suppressed by MMF or CX-5461 (Supplementary Fig. 6n–p), the turnover rate of LEDGF protein (approximately 4 h) is faster than that of most proteins in the proteome, the average turnover rate of which is approximately 20 h[35]. These findings suggest that LEDGF and menin expression is sensitive to perturbed guanine nucleotide biosynthesis and RNA pol I function.

Next, we examined the impact of MMF or CX-5461 on the LEDGF/menin complex formation in AML. Immunoprecipitation of a similar amount of menin protein resulted in an almost completely absence of co-immunoprecipitated LEDGF when AML cells were treated with MMF

or CX-5461 (Fig. 6n). Since LEDGF plays a critical role in tethering the menin/MLL complex to chromatin[6], we assessed menin occupancy to its target genes. We performed ChIP-qPCR of menin in LSCs treated ex vivo with MMF, CX-5461, or VTP-50469, which disrupts the interaction of menin with MLL protein[36]. Consistent with prior studies, VTP-50469 reduced menin binding to its target genes[36]. Treatment with MMF or CX-5461 also reduced menin binding to *Hoxa9* promoter (Fig. 6o). Thus, reduced expression of menin and LEDGF by MMF or CX-5461 impairs the formation of the LEDGF/menin complex and its recruitment to the chromatin.

The LEDGF/menin/MLL-fusion complex is recruited to the target genes via the PWWP domain of LEDGF and the CXXC domain of MLL[37]. A minimal LEDGF/menin/MLL-ENL chimeric protein consisted of the LEDGF PWWP domain, MLL CXXC domain, and ENL (termed PCE) retains the ability to transform HSPCs to AML, similarly to full-length MLL-ENL[37]. Since this chimeric protein bypasses the requirement for LEDGF-menin to recruit the SEC (via ENL), we postulated that this construct may render AML cells resistant to MMF. To test this hypothesis, we ectopically expressed PCE in MOLM-13 cells and exposed the cells to MMF or CX-5461. We found that PCE expression rendered cells resistant to the effects of MMF, CX-5461, or VTP-50469 in inducing myeloid differentiation or suppressing cell proliferation (Fig. 6p,q). Additionally, induction of MLL-AF9 target genes with *Hoxa9-p2a-Meis1-p2a-GFP* construct rendered THP-1 cells resistant to myeloid differentiation induced by MMF (Fig. 6r and Supplementary Fig. 6q). Thus, the main mechanism by which disruption of guanine nucleotide biosynthesis promotes AML differentiation is by reducing chromatin binding of LEDGF/menin. These results establish that disruption of guanine nucleotide or rRNA biosynthesis reduces the expression and complex formation of LEDGF and menin, thereby reducing the chromatin occupancy and target gene expression of the MLL-fusion complex.

### Targeting purine biosynthesis sensitizes AML cells to menin inhibitors
Small molecule inhibitors that disrupt the menin-MLL interaction[8] hold great promise as a therapy for *MLL*r AML. *MLL*r cell lines, including MOLM-13 and MV4-11, were highly sensitive to a menin inhibitor VTP-50469 (Supplementary Fig. 7a, b). Moreover, treatment of LSCs and *MLL*r cell lines resulted in myeloid differentiation upon exposure to a menin inhibitor ziftomenib (Supplementary Fig. 7c, d). The observation that MMF and CX-5461 reduce LEDGF and menin protein, leading

to reduced chromatin occupancy of the menin/MLL-fusion complex raised the possibility that MMF and CX-5461 may sensitize AML cells to menin inhibitors. The combinatorial treatment of VTP-50469 with MMF or CX-5461 in vitro exhibited high synergy in inhibiting the proliferation of *MLL*r MOLM-13 and MV4-11 cells (Fig. 7a–d and Supplementary Fig. 7e). However, no synergistic effects were observed in NB4 cells, a non-*MLL*r cell line (Fig. 7e, f and Supplementary Fig. 7e). Furthermore, we did not find synergistic interactions between MMF and doxorubicin in *MLL*r MOLM-13 and MV4-11 cells (Supplementary Fig. 7f, g and 7e). To determine the efficacy of combined inhibition of guanine nucleotide biosynthesis and menin in vivo, we first evaluated the efficacy of a menin inhibitor ziftomenib in mice transplanted with murine MLL-AF9-induced AML. Ziftomenib treatment for 2 months delayed the onset of leukemogenesis and the frequency of AML cells in the blood in a dose-dependent manner (Fig. 7g and Supplementary Fig. 7h). All mice (9 out of 9 mice) treated with 100 mg/kg/day of ziftomenib became AML free (Fig. 7g). We then treated AML recipient mice with a low dose (12.5 mg/kg/day) of ziftomenib either alone or in combination with 100 mg/kg/day of MMF. We observed a significant reduction of GFP⁺ AML cells in the peripheral blood and prolonged survival of AML mice following treatment with either ziftomenib or MMF alone (Fig. 7h, i). Importantly, the combination treatment of ziftomenib and MMF led to further reduction of circulating AML cells and extension of AML-free survival (Fig. 7h, i). Finally, we generated two *MLL*r-AML PDX models and treated them with MMF (100 mg/kg/day) with or without a low dose of ziftomenib (12.5 mg/kg/day) (Fig. 7k). The combination of MMF and ziftomenib significantly reduced AML burden in the bone marrow compared to single agent treatments (Fig. 7j–l). These data establish that the combinatorial inhibition of purine biosynthesis and menin-MLL interaction strongly suppress *MLL*r AML.

## Discussion

Here we demonstrate that the guanine nucleotide biosynthesis pathway safeguards MLL-AF9 target gene expression by maintaining the LEDGF/menin/MLL-AF9 complex. The reduced binding of the LEDGF/menin/MLL-fusion complex and increased binding of myeloid transcription factors were associated with myeloid differentiation of LSCs and enhanced AML sensitivity to menin inhibition.

This study supports the growing recognition of IMPDH as a promising therapeutic target for cancer. Our finding that inhibition of guanosine production promotes myeloid differentiation, along with a recent report showing that guanosine treatment also induces myeloid differentiation[38], suggests that modulation of the guanosine pathway either through inhibition or overactivation may drive AML differentiation. The dependence on IMPDH has been indicated in a variety of cancer types, including AML[39,40], small cell lung cancer[21], glioblastoma[32], and kidney cancer[41]. Repurposing of MMF, an immunosuppressant used in clinic for almost 30 years with an established safety profile, for cancer therapy is an attractive approach. However, neutropenia, a common side effect observed with MMF, is a concern especially for hematologic malignancies that are associated with increased risk of infection[42]. Our study also identifies RNA pol I inhibition as an alternative to IMPDH inhibition that downregulates the LEDGF/menin/MLL-fusion complex and induces AML differentiation. An RNA pol I inhibitor CX-5461 was well tolerated in clinical trials for hematologic and solid cancers without evidence for neutropenia[43,44]. Development of new compounds that target several steps in rRNA transcription[45] opens new opportunities to sensitize AML to menin inhibition.

The mechanism by which the inhibition of guanine nucleotide biosynthesis or rRNA transcription reduces LEDGF and menin expression requires further investigation. In this context, it is interesting to note the similar effects of purine and pyrimidine biosynthesis on LSC differentiation. A prior study demonstrated that DHODH, an enzyme involved in pyrimidine metabolism, is critical for maintaining LSCs[46]. Inhibition of DHODH reduces protein translation and MLL target gene expression, while increasing the accessibility of chromatin sites that recruit myeloid transcription factors, such as C/EBP and PU1[47]. We speculate that LEDGF is among the most sensitive protein whose expression is affected by purine and pyrimidine biosynthesis and the resulting rRNA transcription blockade.

A recent clinical trial found remarkable single-agent activity of a menin inhibitor revumenib (SNDX-5613) against *MLL*r or *NPM1c*-mutant leukemias[10]. However, acquired resistance developed quickly and somatic mutations in *MEN1* that confer menin inhibitor resistance were detected in approximately 40% of patients[48]. Some patients also developed non-genetic resistance without *MEN1* mutations that downregulated MLL target genes in response to menin inhibition without cellular response[48]. Given that LEDGF is required for menin/MLL to bind chromatin, it is likely that MMF and CX-5461 will be active against *MEN1*-mutant AML but possibly not against the non-genetic resistant clones.

In conclusion, our results establish that the guanine nucleotide biosynthetic pathway maintains leukemogenic gene expression by the MLL-fusion complex. Our study adds guanine nucleotide biosynthesis to the growing list of metabolic pathways or metabolites that impinge on AML gene regulation, such as isocitrate dehydrogenases, α-ketoglutarate, acetyl-CoA, S-adenosylmethionine, and NAD⁺ (refs. 27,49–52). Further exploration of metabolic pathways that impact gene expression should broaden the therapeutic options for leukemias with aberrant gene regulation.

## Methods
### Animal studies
The mice were housed in the Association for Assessment and Accreditation of Laboratory Animal Care International (AAALAC)-accredited, specific pathogen-free animal care facilities at Baylor College of Medicine (BCM). C57BL/6 J (JAX stock: 00664) and B6.SJL-Ptprca Pepcb/BoyJ (CD45.1) mice (JAX stock: 002014) of both sexes at the age of 8-12 weeks were used for murine AML and competitive bone marrow transplantation assays, respectively. *Impdh2ᶠˡ* (JAX stock: 028910) and *TetO-Cas9; M2rtTA* (JAX stock: 029415) were purchased from the Jackson Laboratory. After backcrossing to the C57BL/6 J background, the *Impdh2ᶠˡ* mice were crossed with *Mx1-Cre* transgenic mice (JAX stock: 003556). Tail biopsies at the time of weaning were obtained for mouse genotyping with primers. NOD.Cg-*Prkdcˢᶜⁱᵈ Il2rgᵗᵐ¹ᵂʲˡ* Tg(CMV-IL3,CSF2,KITLG)¹ᴱᵃᵛ/ᴹˡᵒʸˢᶻʲ (NSG-SGM3) (JAX stock: 013062)[53] were purchased from the Jackson laboratory and used for transplanting human AML patent samples. AML progression in mice were assessed daily to identify mice that exhibit signs of distress such as lack of grooming, ruffled fur, hunched posture, lethargy, lack of appetite, difficulty in moving, feeding or drinking. Mice exhibiting these signs of distress were euthanized. Additionally, mice that had more than 50% leukemia cells in the blood, as assessed by flow cytometry, were closely monitored by increasing the frequency of monitoring from once to twice a day. All animal experiments were performed in accordance with protocols approved by the Institutional Animal Care and Use Committee (IACUC) of BCM.

### Poly (I:C), MMF, and ziftomenib treatments
Poly(I:C) (GE Healthcare, Chicago, IL) was suspended in PBS at 200 μg/ml, and mice were injected intraperitoneally with 1 μg/g of body mass every other day for 5 times. MMF was dissolved in 0.5% methycellulose in 0.1% Tween-80 solution and administrated orally at a dose of 100 mg/kg/day[32]. Ziftomenib was dissolved in 20% (w/v) 2-hydroxypropyl-β-cyclodextrin (HP-β-CD, H107, Sigma Aldrich, pH 3.1–3.3), and administered orally at a range of dose of 12.5–100 mg/kg/day. For the combined treatment of MMF and ziftomenib, AML recipient mice were initially administered 100 mg/kg/day daily from day 5

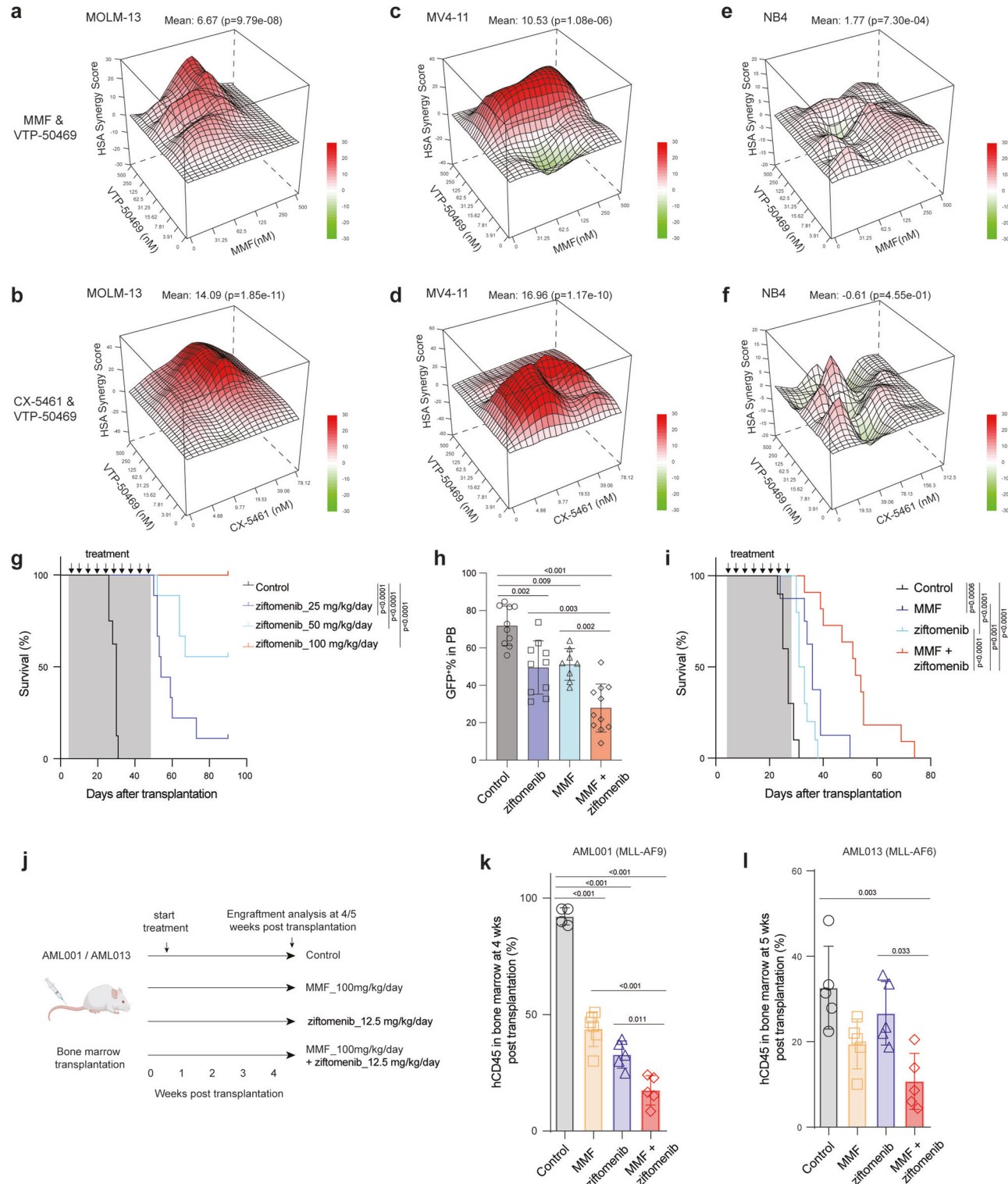

**Fig. 7 | Inhibition of purine biosynthesis sensitizes *MLL*r AML cells to menin inhibitor. a**, **b** HSA synergy score of MMF plus VTP-50469 (**a**) and CX-5461 plus VTP-50469 (**b**) in MOLM-13 cells. A synergy score >10 represents a synergistic effect. **c**, **d** HSA synergy score of MMF plus VTP-50469 (**c**) and CX-50469 (**d**) in MV4-11 cells. **e**, **f** HSA synergy score of MMF plus VTP-50469 (**e**) and CX-5461 plus VTP-50469 (**f**) in NB4 cells. **g** Kaplan-Meier analysis of survival of MLL-AF9-induced AML recipient mice treated with control or ziftomenib (25-100 mg/kg/day) ($n$ = 8-9). **h**, **i** Frequency of GFP⁺ AML cells in PB (**h**) and survival analysis (**i**) of AML mice treated with control ($n$ = 10), MMF (100 mg/kg/day) ($n$ = 8), ziftomenib (12.5 mg/kg/day) ($n$ = 10), or a combination of MMF and

ziftomenib ($n$ = 11 mice). **j** Schematic flowchart created with Biorender depicting the AML PDX study with MMF (100 mg/kg/day) and ziftomenib (12.5 mg/kg/day) treatments. **k**, **l** Frequency of hCD45+ AML cells in bone marrow of mice transplanted with AML001 (**k**) (Control $n$ = 4, MMF $n$ = 6, ziftomenib $n$ = 5, and combination $n$ = 5 mice) and AML013 PDX samples (**l**) ($n$ = 5 mice per group). All data are represented as mean ± SD. p values in this figure were calculated by permutation tests (**a**–**f**) and ANOVA with multiple comparisons analysis using Bonferroni correction post hoc analyses (**h**, **k**, and **l**), or log-rank test (**g** and **i**). Source data are provided as a Source Data file.

post-transplantation for the first week. Following this, the regimen was adjusted to every other day, according to a previous protocol[54]. Mice were treated daily with ziftomenib (12.5 mg/kg/day for both murine and PDX models) via oral gavage throughout the experiment.

## Human subjects

Human AML patient cells were collected under LAB01-473 protocol (approved by the MD Anderson IRB committee) or H-3342 (approved by the Texas Children's Hospital (TCH) IRB committee) from patients who consented to participate in the MD Anderson, TCH, or Children's Oncology Group (COG) tissue banking protocols. These AML patient cells were expanded in immunodeficient murine models, and cultured in Iscove's Modified Dulbecco's Medium (IMDM) supplemented with human stem cell factor (SCF, 100 ng/ml), granulocyte colony-stimulating factor (G-CSF, 20 ng/ml), interleukin 3 (IL-3, 20 ng/ml), fms related receptor tyrosine kinase 3 ligand (FLT3 Ligand, 50 ng/ml) (all from Peprotech), 15% FBS, and 2-mercaptoethanol (100 mM) for 24 h to recover. After this recovery period, the leukemia cells were treated with a guanine biosynthesis inhibitor and differentiation was assessed after 6 days of treatment. All the procedures involving the use of human samples were approved by the BCM Institutional Review Board (IRB) committee under protocol H-43877. Characteristics of each human AML specimen used for analysis are included in Supplementary Data 3[55].

## Cell lines and reagents

The 293 T cells were purchased from the American Type Culture Collection (#CRL-3216, ATCC) and cultured in Dulbecco's modified Eagle medium (DMEM). The leukemia cell lines MOLM-13, THP-1, HL-60, MV4-11, NOMO1, NB4, K562, Kasumi1, KG1, OCI-AML2, and OCI-AML3 (from M. Goodell, R. Rau, D. Lacorazza, or ATCC) were cultured in RPMI-1640. All cell lines were cultured in media supplemented with 10% (volume/volume; v/v) FBS and 1% (v/v) penicillin/streptomycin at 37 °C in an incubator with a humidified atmosphere of 5% $CO_2$ and 20% $O_2$. The cell lines were routinely tested free of mycoplasma contamination and authenticated by Cytogenetic and Cell Authentication Core at MD Anderson Cancer Center. L-GMP, referred to as LSCs, were sorted from MLL-AF9-driven leukemia mice and cultured in RPMI-1640 medium supplemented with 20% FBS and mouse IL3 (10 ng/ml) (Peprotech). All cell lines were initially cultured in a 10-cm dish and subsequently sub-cultured in a 24-well or 96-well plate for the competitive growth assay and viability assay, respectively. Cells were treated with mycophenolate mofetil (MMF, #S1501, Sellechem), mycophenolic acid (MPA, #M5255, Sigma Aldrich), 6-mercaptopurine monohydrate (6-MP, #852678, Sigma Aldrich), Mizoribine (#M3047, Sigma Aldrich), Tiazofurin (#HY-114570, MedChemExpress), CX-5461 (#S2684, Selleckchem), BMH-21 (#SML1183, Sigma Aldrich), VTP-50469 (#HY-114162, MedChemExpress) or SNDX-5613 (#HY-136175, MedChemExpress) for 2-6 days, or as otherwise indicated. Ziftomenib (KO-539) was a gift from Kura Oncology (San Diego). All the compounds were dissolved in DMSO unless otherwise indicated by the instructions.

## Flow cytometry

Bone marrow cells were either flushed from the long bones (tibias and femurs) or isolated by crushing the long bones (tibias and femurs), pelvic bones, vertebrae, and sternum with mortar and pestle in 1x Hank's buffered salt solution (HBSS) without calcium and magnesium, supplemented with 2% heat-inactivated bovine serum (ThermoFisher). Splenocytes or thymocytes were dissociated by mashing the entire spleen or thymus between frosted slides. Peripheral blood was incubated with ACK buffer (150 mM of Ammonium chloride, 10 mM of Potassium bicarbonate, and 0.1 mM of EDTA) to lyse red blood cells. Cells were triturated and filtered through a nylon screen (100 µm,

Sefar America) or a 48 µm cell strainer (ThermoFisher) to obtain a single-cell suspension. For isolation of HSCs (lineage⁻Sca-1⁺c-kit⁺(LSK) CD150⁺CD48⁻/ˡᵒʷ), MPPs (LSKCD150⁻CD48⁻/ˡᵒʷ), HPC2 (LSKCD150⁺ CD48⁺), and HPC1 (LSKCD150⁻CD48⁺), bone marrow cells were incubated with PE-Cy5-conjugated anti-CD150 (TC15-12F12.2; BioLegend), PE-Cy7-conjugated anti-CD48 (HM48-1), APC-conjugated anti-Sca-1 (Ly6A/E; D7), and biotin-conjugated anti-c-kit (2B8) antibody, in addition to antibodies against the following PE-conjugated lineage markers: Ter119 (TER-119), B220 (6B2), Gr-1 (8C5), CD2 (RM2-5), CD3 (KT31.1), and CD8 (53-6.7). In some experiments, Flk2 (A2F10) was included in the panel to identify MPP2 (LSKFlk2⁻CD150⁺CD48⁺), MPP3 (LSKFlk⁻CD150⁻CD48⁺) and MPP4 (LSKFlk2⁺CD150⁻CD48⁺). Biotin-conjugated antibodies were visualized using streptavidin-conjugated APC-Cy7.

GMP (Lin⁻Sca1⁻c-kit⁺CD34⁺CD16/32⁺), common myeloid progenitors (CMP, Lin⁻Sca1⁻c-kit⁺CD34⁺CD16/32⁻/ˡᵒʷ), megakaryocyte erythroid progenitors (MEP, Lin⁻Sca1⁻c-kit⁺CD34⁻/ˡᵒʷCD16/32⁻/ˡᵒʷ), and common lymphoid progenitors (CLP, Lin⁻Sca1ˡᵒʷc-kit⁺IL7R⁺Flk2⁺) were isolated based on the established cell surface markers.

To analyze T-cell development in the thymus, Lin⁻ (Mac1, Gr1, B220, and Ter119) thymocytes were separated based on CD4 (GK1.5), CD8 (53-6.7), CD44 (IM7) and CD25 (PC61.5) to distinguish various stages of T-cell maturation. The stages of T-cell maturation studied are DN1 (CD4⁻CD8⁻CD25⁻CD44ʰⁱᵍʰ), DN2 (CD4⁻CD8⁻CD25⁺CD44ʰⁱᵍʰ), DN3 (CD4⁻CD8⁻CD25⁺CD44ˡᵒʷ), DN4 (CD4⁻CD8⁻CD25⁻CD44⁻), double positive (CD4⁺CD8⁺), and mature CD4⁺ or CD8⁺ single positive cells.

For HSPC isolation, bone marrow cells were pre-enriched by selecting c-kit⁺ cells using paramagnetic microbeads and autoMACS (Miltenyi Biotec). AML cells from the bone marrow or spleens were incubated with antibodies against Sca-1, c-kit, CD16/32, CD34, and lineage markers (CD2, CD3, CD8, Gr1, Ter119, and B220) to identify LSCs (GFP⁺ GMP, also known as leukemic-GMP) as described previously[27,56]. Bulk AML cells were identified as GFP⁺ cells. Myeloid differentiation of AML cells was determined by staining with antibodies against CD11b, Gr-1, CD14, CD33, and CD66b. Nonviable cells were excluded from sorts and analyses using the viability dye 4′,6-diamidino-2-phenylindole (DAPI) (1 µg/ml). Unless otherwise noted, antibodies were obtained from BioLegend, BD Biosciences, or ThermoFisher. Flow cytometry was performed with FACSAria II, FACSCanto II, LSR II, or LSRFortessa flow cytometers (BD Biosciences). All flow cytometry gating strategy are shown in Supplementary Fig. 8.

## Metabolomics profiling

The bone marrow from AML recipient mice and sex-matched C57BL/6 mice were used for sorting $10^5$ LSCs, GFP⁺ AML cells, normal GMP and CD45⁺ WBM cells, respectively, with 7 replicates per group. Cells were kept on ice during the entire isolation, staining and sorting procedure. After staining with respective antibodies, cells were suspended in 1xHBSS plus 0.2% BSA. Aria III cytometer was cleaned with windex/ethanol/deionized water in sequence with 0.5x PBS as the sheath fluid right before sorting. The sorted cells were centrifuged at 1,500 rpm for 5 minutes and washed once with ice-cold 1x HBSS. The solution was aspirated, leaving approximately 20 µl behind. 60 µl of ice-cold acetonitrile was added and vortexed for 1 minute. The cell lysate was frozen at -80°C for 30 minutes, thawed on ice and cell debris pelleted at 13,000 rpm for 15 minutes to collect the supernatant. The metabolomics analysis was performed with a QExactive HF-X hybrid quadrupole orbitrap high-resolution mass spectrometer (ThermoFisher) as previously described[57,58]. Chromatogram peak areas from a targeted metabolite list were integrated using Tracefinder.

For nucleotide metabolomics, cultured LSCs were treated with 0.5 µM MMF and/or 100 µM guanosine for 2 h, with 5 replicates per group. After treatment, 5 ×10⁵ cells were collected, centrifuged at 1,500 rpm for 5 minutes, and washed twice with PBS. The cells were dissolved in

100 μl of acetonitrile: methanol: ddH₂O (40:40:20) plus 0.1 M formic acid solution with vortexing. The cell lysate was neutralized with ammonium hydroxide, frozen at -80°C for 30 minutes, pelleted, and the supernatant collected for LC/MS.

## Stable isotope tracing assay by IC-MS

The stable isotopes $^{13}C_6$-glucose and amide-$^{15}$N-glutamine were purchased from Cambridge Isotope Laboratories. The incorporation of glucose carbon and glutamine nitrogen into purine nucleotides was determined by ion chromatography-mass spectrometry (IC-MS). LSCs and non-LSCs (CD34⁻ counterpart of LSCs) were isolated from MLL-AF9-induced AML mice and incubated in RPMI1640 medium supplemented with 20% dialyzed FBS, 10 ng/ml mouse IL3, 1% penicillin/streptomycin, and 2 mg/ml $^{13}C_6$-glucose or 0.3 mg/ml amide-$^{15}$N-glutamine for 1 and 4 h, immediately followed by an ice-cold PBS wash to remove residual medium components. Metabolites were extracted from the cells using 1 mL ice-cold 80/20 (v/v) methanol/water containing 0.1% ammonium hydroxide. Then, the samples were centrifuged at 17,000 g for 5 minutes at 4°C, and the resulting supernatants were transferred to clean tubes and evaporated under nitrogen. The samples were reconstituted in deionized water, and 5 μl of each sample was injected into a Dionex ICS-6000+ capillary ion chromatography (IC) system (ThermoFisher) equipped with a Thermo IonPac AS11 column (4 μm particle size, 250 × 2 mm) for the analysis of polar metabolites. The IC system operated at a flow rate of 360 μl/min (at 30°C), and the gradient conditions were as follows: started with an initial 1 mM KOH, increased to 35 mM at 25 minutes, then to 99 mM at 39 minutes, held 99 mM for 10 minutes. The total run time was 50 minutes. To improve sensitivity through desolvation, methanol was introduced using an external pump and combined with the eluent via a low dead-volume mixing tee. Data were acquired using a Thermo Orbitrap IQ-X Tribrid Mass Spectrometer under ESI negative mode at a resolution of 240,000.

Thermo Trace Finder 5.1 software was used to analyze the $^{13}C_6$-glucose and amide-$^{15}$N-glutamine labeling. The fractional abundance of each isotopologue was determined by calculating the peak area of the corresponding isotopologue and normalized it by the sum of the peak areas of all isotopologues, as described by Du et al.[59]. Mass distribution vector (MDV) represents the fractional abundance of measured ions for each isotopolog after correction for natural abundance and isotopic impurity. Isotopic enrichment (IE) is defined as

$$\sum_{i=1}^{N} i \cdot Mi$$

Here, $M_i$ is the MDV, and $N$ signifies the total number of carbon or nitrogen atoms in a molecule. The value of IE denotes the labeling ratio, serving as an indicator of the metabolic rate, where higher values represent a greater metabolic rate for producing a specific metabolite.

## Analysis of polar metabolites by IC-MS

To assess the predominant utilization of guanosine metabolites for rRNA transcription in LSCs, LSCs were treated with 0.5 μM of MMF, 0.5 μM CX-5461 or a combination of 0.5 μM MMF and 0.5 μM CX-5461 (1-hour pretreatment) for 1.5 h. 1 x 10⁶ LSCs were collected, centrifuged at 1500 rpm for 5 minutes, and washed twice with PBS to remove the culture medium. Metabolites were extracted with ice-cold 0.1% ammonium hydroxide in 80/20 (v/v) methanol/water and centrifuged at 17,000 g for 5 minutes. The supernatant was evaporated to dry under nitrogen and reconstituted in deionized water. Approximately 10 μl of each sample was injected into the Dionex ICS-5000+ Capillary IC System (ThermoFisher) equipped with the Thermo IonPac AS11 column. IC mobile phase A was water and mobile phase B was 100 mM KOH. The autosampler tray was chilled to 4°C. The mobile phase flow rate was 360 μl/min (at 30°C), and the gradient elution program was:

0-5 minutes, 1%; 5–25 minutes, 1–35%; 25–39 minutes, 35–99%; 39–49 minutes, 99-100% mobile phase B. Data was acquired with Orbitrap Fusion Tribrid Mass Spectrometer (ThermoFisher) and imported to Trace Finder 4.1 software (ThermoFisher) for analysis. The relative abundance of each metabolite was normalized by total peak intensity.

## Drug response experiments

To examine cell viability with single drug treatment or genetic manipulation, 0.2-1×10⁴ cells were seeded in a final volume of 100 μl per well in flat bottom 96-well plates. The concentration of DMSO in the incubation mixtures and solvent control wells was less than 0.5% (v/v). Cell viability was measured with CellTiter 96 Aqueous One Solution Cell Proliferation Assay (Promega) according to the manufacturer's instructions after 4 days of treatment. Specifically, 20 μl of the Aqueous One Solution Reagent was added into each well of the plates, followed by incubation at 37 °C for 1-4 h. The absorbance was recorded at 490 nm using Plate reader Infinite M200 Pro (Tecan). Cell viability rates were normalized to corresponding DMSO or media control cells. The drug combination landscapes (5*8) were built by Bioconductor "synergyfinder" package or web application (https://synergyfinder.org) with ZIP, Loewe, HSA and Bliss models using multiple-ray design to assess the potential synergy of drug combinations[60,61]. Each experiment was performed at least three times independently.

## Retroviral transduction

293 T cells cultured in a 10-cm plate were transiently transfected with MSCV vectors (10 μg) along with pCL-Eco (5 μg, Addgene #12371) plasmid in Opti-MEM using a PEI transfection protocol. Sodium butyrate was added at a concentration of 5 μM the day after transfection to enhance retrovirus production. The retrovirus in culture medium was collected 48 h after transfection, and incubated with 80 mg/ml PEG-6000, 100 mM NaCl, and 10 mM Hepes-NaOH (pH 7.4) overnight before centrifuging at 1500 g for 45 minutes at 4 °C. The concentrated retrovirus was dissolved in the mouse hematopoietic stem and progenitor cells (HSPCs) culture media X-Vivo 15 (Lonza), supplemented with murine thrombopoietin (TPO, 50 ng/ml), SCF (10 ng/ml), IL-3 (10 ng/ml), IL-6 (10 ng/ml) (all from Peprotech), 1% FBS and 1% penicillin/streptomycin. HSPCs (lineage⁻Sca-1⁺c-kit⁺ cells) sorted from mice were incubated for 18-24 h in HSPCs culture media. Subsequently, the cells were spin-infected with concentrated retrovirus supplemented with polybrene (8 μg/ml) in retronectin (Clontech) coated plates at 600 g for 30 minutes at room temperature. For primary AML transplantation, approximately 50,000 HSPCs transduced with MLL-AF9 were mixed with 100,000 supporting C57BL/6 BM cells and transplanted into lethally irradiated mice (2 doses of 5 Gy with at least 2 h interval).

## Syngeneic transplantation

For secondary or limiting dilution BM transplantation assay (LDA), different numbers of GFP⁺ AML cells from the BM of primary recipient mice were transplanted into sub-lethally (650 cGy, single dose) irradiated recipient mice. A total of 10,000-20,000 AML cells were transplanted into secondary recipient mice for MMF and KO-539 treatment.

For whole BM competitive transplantation assays, 10⁶ BM cells from *Impdh2*^fl/fl^, *Mx1-Cre; Impdh2*^+/fl^ or *Mx1-Cre; Impdh2*^fl/fl^ mice were mixed with 10⁶ BM cells from CD45.1 mice and transplanted retro-orbitally into lethally irradiated CD45.1 recipient mice. Poly (I:C) was administrated on day 5 post-transplantation to delete *Impdh2*. Peripheral blood was collected from these recipient mice at the indicated time-point after transplantation and donor chimerism was measured in Mac-1⁺/Gr1⁺ myeloid cells, B220⁺ B-cells, and CD3⁺ T-cells within

CD45.2$^+$ cells by flow cytometry. We analyzed these mice for up to 6 months after transplantation.

## Complete blood count (CBC) analysis

Peripheral blood samples (20-30 µl) were collected in MiniCollect capillary blood collection tubes (#574506, Greiner Bio-One) and then analyzed by the Scil Vet Abc Plus hematology analyzer (Scilvet) according to the manufacturer's instructions.

## In vitro colony forming assay

For the colony-forming assay,100 LSCs were seeded into each well of a 6-well plate containing 2 ml of methylcellulose medium (M3234, Stem Cell Technologies) supplemented with 10 ng/ml of mouse IL-3 after treatment with 125 nM and 250 nM of MMF for 12 h. Colonies were quantified after 7-10 days of culture, and images of the 6-well plate were captured using EVOS FL Auto Imaging System (ThermoFisher). Individual colony images were acquired using the Axiovert 25 Inverted Phase Contrast Microscope (Zeiss). Then, the colonies were assessed for myeloid differentiation (CD11b-PE) and re-plated with an additional two rounds to determine colony number.

## Cloning, lentivirus production, and Cas9 functionality assay

The cloning of individual sgRNA into lentiviral vector pKLV2-U6gRNA5(BbsI)-PGKpuro2ABFP-W (Addgene #67974) was performed with a BbsI enzyme (NEB). Sanger sequencing was used to verify from a single clone. All lentiviral constructs, including Cas9 (Addgene #68343), Cas9 reporter (pKLV2-U6gRNA5(gBFP)-PGKBFP2AmCherry-W), pTY-EF1a-Hoxa9-p2a-Meis1-GFP (Addgene #61738), and the sgRNAs constructs were packaged by transient transfection of 293 T cells with psPAX2 (Addgene #12260) and pMD2.G (Addgene #12259) in a 5.4 µg:5.4 µg:1.2 µg ratio in a 10-cm dish by using PEI solution. The lentivirus-containing supernatant was pooled from day 2 and day 3 after transfection. It was then passed through a 0.45 µm filter, precipitated with PEG-6000 solution. After incubation overnight at 4 °C, the mixed solution was centrifuged at 1,500 g for 45 minutes at 4 °C. The resulting pellet was suspended in a culture medium and stored in small aliquots at −80 °C.

The coding sequence of MYC was amplified from a MYC estrogen receptor (ER) fusion construct (kindly provided by Dr. Trey Westbrook at Baylor College of Medicine) and subsequently cloned into a pMIG-FLAG vector. pMIG-3xTy1-PCE was generated by cloning the PCE sequence from pMSCV-neo-PCE plasmid (generously provided by Dr. Akihiko Yokoyama[37]) into a pMIG-3Ty1. All new constructs were verified by Sanger sequencing.

Human leukemia cell lines were transduced with the Cas9 lentivirus and then treated with blasticidin (10 µg/ml) 3 days after transduction to select for Cas9-expressing cells. The functionality of Cas9 in these leukemia cells were assessed by transduction with pKLV2-U6gRNA5(gBFP)-PGKBFP2AmCherry-W that expresses BFP, mCherry, and an sgRNA targeting BFP expression. The ratio of mCherry$^+$ and BFP$^+$mCherry$^+$ cells at 3-4 days after transduction was analyzed by flow cytometry.

## CRISPR/Cas9-mediated competitive growth assay in vitro

Deletion of *IMPDH2* and other genes involved in the purine biosynthesis pathway was performed with CRISPR/Cas9-mediated gene editing as previously described[27,56]. Cas9-expressing leukemia cell lines were transduced with individual sgRNA in the presence of polybrene (0.4 µg/ml) to achieve a transduction efficiency of 30-50% at 4 days after transduction, as assessed by BFP expression using flow cytometry (BD). This served as a reference for the percentage of sgRNA/BFP$^+$ cells. The relative competitive growth in sgRNA/BFP$^+$-expressing cells versus sgRNA/BFP$^-$ cells was monitored over 3 weeks by flow cytometry and normalized to the reference taken on day 4 after transduction. Propidium iodide (PI) was used to gate live cells.

## Inducible CRISPR/Cas9 in murine leukemia cells

HSPCs sorted from *TetO-Cas9; M2rtTA* mice (JAX stock: 029415) were transduced with MLL-AF9 retrovirus and then transplanted into lethally irradiated recipients to generate a murine AML model. When the GFP$^+$ leukemia cells in the peripheral blood reached 80-90%, whole bone marrow cells were collected from primary AML mice and cultured in RPMI 1640 medium supplemented with 20% FBS, 10 ng/ml murine IL-3, and 1% penicillin/streptomycin. After 12 h of culture, AML cells were transduced with BFP-expressing sgRNA lentivirus targeting *Impdh1*, *Impdh2* or control *Rosa26* locus, and treated with 1 µg/ml of doxycycline for 48 h to induce Cas9 expression. The percentage of sgRNA/BFP$^+$ leukemia cells was assessed by flow cytometry over 4 days of culture.

## CRISPR/Cas9 knock-in assay in human AML cells

To generate the THP-1 cells expressing NPM1-mNeonGreen, electroporation was performed using a mixture of plasmids carrying NPM1-sgRNA (eSpCas9(1.1)_No_FLAG_NPM1_G6, #178091, Addgene) and donor template (NPM1_mNeonGreen_Donor, #178093, Addgene), as previously described[62]. Briefly, $1 \times 10^5$ THP-1 cells were washed with PBS, resuspended in 10 µl of buffer R from the Neon Transfection System 10 µL kit (#MPK1096, ThermoFisher) and mixed with 1 µl of the NPM1-sgRNA (100 ng) and 1 µl of the donor template (250 ng). The cell/sgRNA/donor mixture (10 µl out of a total of 12 µl) was electroporated using the Neon Transfection System (ThermoFisher) with the condition of 1350 V, 20 ms, 1 pulse. 3 days after electroporation, the THP-1 cells expressing NeonGreen were sorted twice using flow cytometry and then treated with the indicated compounds.

## Immunoblotting

Western blotting was performed to analyze protein expression as previously described[63]. Briefly, AML cells were incubated in 10% (w/v) trichloroacetic acid (TCA) on ice for 10 minutes before centrifugation. The resulting pellets were washed with acetone twice, air dried, and dissolved in solubilization buffer along with LDS loading buffer. After denaturing at 70°C for 10 minutes, proteins were separated on a Bis-Tris polyacrylamide gel and transferred to a PVDF membrane (Millipore). Antibodies used for immunoblotting are anti-IMPDH1 (1:1000, #28420, Cell Signaling Technology), anti-IMPDH2 (1:1000, #36281, Cell Signaling Technology), anti-FLAG (1:1000, #14793, Cell Signaling Technology), anti-NF-YA (1:1000, sc-17753, Santa Cruz), anti-menin (1:1000, #A300-105A, ThermoFisher), anti-LEDGF(1:1000, #2088, Cell Signaling Technology), anti-NF-Y1 (1:1000, #SAB4800032, Sigma Aldrich), anti-MYC (1:1000, #9402, Cell Signaling Technology), anti-puromycin (1:1000, #MABE343, Sigma Aldrich),anti-Ty1 (1:1000, #SAB4800032, Sigma Aldrich), anti-GAPDH (1:5000, 12118, Cell Signaling Technology), and anti-β-actin (1:5000, A1978, Sigma Aldrich). After incubation with appropriate anti-rabbit or anti-mouse horseradish peroxidase (HRP)-conjugated secondary antibodies (#7074 and #7076, 1:5000, Cell Signaling Technology), we detected and acquired signals with a SuperSignal West Pico PLUS Chemiluminescent Substrate (#34580, ThermoFisher) and a KODAK X-OMAT 2000 Processor (Kodak) or ChemiDoc MP Imaging System (Bio-Rad).

## Chromatin immunoprecipitation

Cells treated with MMF were crosslinked in 1% formaldehyde for 10 minutes and quenched by 250 mM glycine for 5 minutes at room temperature with rotation at 15 rpm. Cells were extracted in 50 mM Tris-HCl (pH 8.0), 100 mM NaCl, 5 mM EDTA, and 1% SDS for 10 minutes at room temperature, followed by nuclei precipitation by centrifugation at 10,000 g. The nuclei were then lysed in 66 mM Tris-HCl pH 8.0, 100 mM NaCl, 5 mM EDTA, 0.5% SDS, 1.7% Triton X-100 and sheared with Bioruptor Pico Sonicator (Diagenode) to obtain chromatin fragments of 200-400 base-pairs. The DNA fragments were immunoprecipitated with anti-menin antibody (#A300-105A,

ThermoFisher) and protein A magnetic beads (ThermoFisher), followed by quantitative real-time PCR.

## Wright-Giemsa staining

LSCs were treated with different concentrations of purine biosynthetic inhibitors, including MMF, MPA, and 6-MP, for 24 h. Approximately 25,000 cells were spun onto glass slides at 1,500 rpm for 5 minutes with StatSpin Cytofuge (Beckman Coulter). The slides were dried and stained with Wright-Giemsa solution according to the manufacturer's instruction (Sigma Aldrich). Slides were analyzed on a DMI6000 fluorescence microscope (Leica).

## Cell cycle and apoptosis analysis

LSCs were treated with a range of 250 nM to 1000 nM of MMF or other compounds for 24 h with or without supplementation of guanosine (100 µM). To assess the cell cycle, the cells were then incubated with 10 µM BrdU (Sigma) for 1 hour. The BrdU pulsed cells were fixed, permeabilized, and treated with DNase I to expose incorporated BrdU. The cells were then incubated with APC-conjugated anti-BrdU and 7-AAD (BD, cat# 552598). To assess apoptosis, cells were incubated with FITC Annexin V and PI in 1x Annexin V binding buffer for 15 minutes at room temperature in the dark (BD, cat# 556547). Both cell cycle and apoptosis analysis were analyzed with flow cytometry.

## Puromycin incorporation assay

To measure protein translation, LSCs were isolated from MLL-AF9 AML mice and then treated with MMF for about 20 h, followed by incubation with 1 µg/ml puromycin at 37 °C for 1 hour. The cells were collected and washed once with 1x PSB before lysis for immunoblotting with an anti-puromycin antibody. Alternatively, puromycin-incorporated cells were examined via flow cytometry. In brief, the cells were fixed in BD Cytofix/Cytoperm buffer for 15 minutes, permeabilized by Cytoperm Permeabiliizatoin Buffer Plus for 10 minutes, and then re-fixed in Cytofix/Cytoperm buffer for 5 minutes, all on ice (BD Biosciences). After washing with 1 X BD Perm/Wash buffer, the cells were incubated with anti-puromycin antibody (1:1000, MABE343, Sigma Aldrich) for 1 hour, followed by staining with FITC-conjugated goat anti-mouse IgG H&L (1:2000, ab6785, Abcam) for 30 minutes at room temperature. Cells were stained with DAPI and analyzed by flow cytometry.

## Phagocytosis assay

Group B *Streptococcus* COH1 (ATCC BAA-1176) was grown in Todd-Hewitt broth (THB) supplemented with 5 µg/mL erythromycin and labeled with 1 µM SYTO deep red nucleic acid stain (ThermoFisher, Cat# S34900) for 30 minutes at 37 °C. Alternatively, Group B *Streptococcus* COH1 expressing pDestErm-GFP[64] was used. L-GMPs were treated with 500 nM MMF for 24 to 48 h and then incubated with the deep red nucleic acid stain-labeled COH1 bacteria for 1 hour at 37 °C with gentle shaking. After washing, the percentage of cells phagocytosed COH1 bacteria was analyzed by flow cytometry. The images of phagocytosing cells were taken with a Leica DMI6000 fluorescence microscope and a LAS software after cells were stained with Hoechst 33342 (ThermoFisher).

## RNA-seq and data analysis

Freshly sorted LSCs were treated with 500 nM MMF, or 250 nM CX-5461, with or without 100 µM guanosine for 24 hrs. The cells were lysed in Trizol (ThermoFisher). RNA samples were treated with DNase and purified using a RNeasy MinElute Cleanup kit (QIAGEN). Complementary DNA (cDNA) libraries were amplified according to a modified SMART-seq2 protocol[65]. Approximately 250 pg of the cDNA was fragmented and barcoded for sequencing using a Nextera XT DNA Library Preparation Kit (Illumina). RNA-seq libraries were sequenced

on a NextSeq platform. The raw sequencing data were converted to fastq files and mapped to mm10 using the Spliced Transcripts Alignment to a Reference (STAR) algorithm. The data processing, normalization, and differential expression were analyzed by DESeq2. Geno Ontology (GO) analysis was performed using DAVID.

## ATAC-seq and data analysis

ATAC-seq was performed as previously described[66]. LSCs were treated with 500 nM MMF for 24 hrs, and then 10,000 cells were sorted into 0.01% BSA in PBS in LoBind tubes. After incubation with ATAC lysis buffer (10 mM Tris, pH 7.4, 10 mM NaCl, 3 mM MgCl₂, and 0.1% NP-40), the cells were tagmented with 0.5 µl of Tn5 (0.885 mg/ml) and incubated at 37°C for 30 minutes. The DNA fragments were purified by a DNA purification kit (D4002, Zymo) and amplified with NEBNext Q5 Hot Start HiFi PCR Master Mix (M0543S, NEB). The ATAC-seq library was purified by solid-phase reversible immobilization (SPRI) beads and sequenced on the NextSeq platform. Raw reads were mapped to mm10 using Bowtie2. Peaks were called on each sample using MACS2, and the differential peaks were analyzed using PePr. The center of each location was used as the reference point and was extended ±5 kb from the center point. Motif analysis was conducted using HOMER.

## Real-time qPCR

Total RNAs were extracted with Trizol (ThermoFisher) and reversed transcribed to cDNA using Superscript Vilo Master Mix (ThermoFisher) according to the manufacturer's instructions. Real-time PCR reaction with TB Green Premix Ex Taq II (Tli RNase H Plus) (Takara) or SsoAdvanced Universal SYBR Green Supermix (Bio-Rad) was conducted on the ViiA7 (Applied Biosystems) or CXF96 (Bio-Rad) real-time PCR system, respectively. Triplicate analyses were performed using the $2^{-\Delta\Delta C(T)}$ method, with β-actin/*GAPDH* serving as the housekeeping gene. The primer sequences are described in supplemental Supplementary Data 8.

## ChIP-Atlas and DepMap analysis

Transcription factors binding in the promoter regions (-250bp to +1000 bp of start codon) of purine biosynthesis genes in murine hematopoietic cells were downloaded from the ChIP-Atlas dataset (https://chip-atlas.org/). The dependency scores for the purine biosynthesis genes were downloaded from the DepMap dataset (https://depmap.org/portal/) and further analyzed based on subsets of AML and other types of cancers.

## Statistics and reproducibility

The data are presented as means ± standard deviation (s.d.) of at least three biologically independent experiments or samples. Statistical significance (*p* value) was calculated with an unpaired, two-tailed Student's *t* test, one-way ANOVA using Bonferroni correction or Dunnett's test, Pearson/Spearman correlation test or Log-rank (Mantel-Cox) test by GraphPad Prism 9 or StataSE. ns, not significant. Specifically, for the comparative analysis between two independent groups, we utilized the independent samples unpaired, two-tailed Student's *t* test to assess differences. In scenarios requiring comparisons across multiple groups, the ANOVA test was utilized to ascertain the existence of discrepancies among the groups. To delve into specific differences between individual groups, we implemented a Bonferroni correction, derived from the ANOVA test results, to control the Type I error rate. When assessing multiple treatment groups in comparison to a single control group, Dunnett's test was the preferred method. To evaluate the presence of a correlation between two groups, we initially assessed the normality of the data within each group. Upon confirmation of joint normality across both groups, the Pearson correlation test was applied. Otherwise, the Spearman correlation test was employed as an alternative. The Mantel-Cox test was utilized to compare the survival

outcomes between two groups. MMF and chemotherapy treatment of AML mice was performed once, and all the remaining experiments were independently repeated more than two times with similar results. No methods were used to pre-determine sample size, and the investigators were not blinded to allocation during experiments and outcome assessment.

## Reporting summary

Further information on research design is available in the Nature Portfolio Reporting Summary linked to this article.

## Data availability

All sequencing data used in this study is available from GEO (Accession GSE263344 and GSE263345). Metabolic raw data were deposited to MassIVF (Accession MSV000095343) Source data are provided with this paper. gRNA sequences used in this study are listed in Supplementary Data 8. *Impdh2ᶠˡ* and *TetO-Cas9; M2rtTA* mice are available from JAX. Source data are provided with this paper.

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

## Acknowledgements

We thank Kura Oncology for the generous gift of ziftomenib. D.N. was supported by the National Institutes of Health (NIH, R01CA193235, R01CA255813, and P01CA265748). D.N. and M.A.H.S. were supported by the NIH (R01HL165145). M.A. was supported by the NIH (R01DK125713), Cancer Prevention and Research Institute of Texas (CPRIT) (RR180007), and Alex's Lemonade Stand Foundation 'A' award. X.S. was supported by Four Diamonds (169077) and a Special Fellow Award from the Leukemia and Lymphoma Society. K.T. and D.N. are Scholars of the Leukemia and Lymphoma Society. Y.K. was supported by the Uehara Memorial Foundation. J.T. was supported by the NIH (F31HL164097). S.H. was supported by the NIH (R01CA260729 and R01CA264932). K.T. was supported by the MD Anderson Physician Scientist Program, the MD Anderson AML/MDS Moonshot Program, and the NIH (P50CA100632 and R01CA237291). Development of PDX models utilized here was supported by the Leukemia and Lymphoma Society, the Target Pediatric AML/the Children's Oncology Group Foundation, CPRIT (RP170691 and RP220646), and gifts of funding from the Turn it Gold Fund (A.M.S.). Flow-cytometry was partially supported by the CPRIT Core Facility Support Award (RP180672) and the NIH (CA125123 and RR024574). Sequencing and PDX development were partially supported by the NIH (P30CA125123).

## Author contributions

X.S performed most of the experiments with assistance from M.L., J.T., Y.L., J.Z, Y.Z., M.F., Y.K., Y.H., J.J.T., X.L., S.M., Q.D., L.T., P.L.L., S.H., and M.A. performed the metabolomics analysis. Z.L. performed bioinformatics analysis under the supervision of M.A.H.S. K.A.P. provided reagents and input on phagocytosis assays. A.M.S. provided some PDX samples and input on PDX experiments. K.T. and G.C.I. provided expertise on menin inhibitors. X.S. and D.N. designed experiments and wrote the manuscript. D.N. obtained the research fund and administered the study. All authors commented on the manuscript.

## Competing interests

K.T. received consultancy fees from Symbio Pharmaceuticals, honoraria from Mission Bio, Otsuka, and Illumina, and research funding from Onconova, ASTEX, and Jazz pharmaceuticals. G.C.I. received consultancy or advisory role fees from Novartis, Kura Oncology, Syndax Pharmaceuticals, Abbvie, Sanofi and NuProbe and received research funding from Celgene, Novartis, Kura Oncology, Syndax Pharmaceuticals, Merck, Cullinan Oncology, Astex, NuProbe, and received research funding from Break Through Cancer. The remaining authors declare no competing interests.

## Additional information

[1]Division of Pediatric Hematology/Oncology, Department of Pediatrics, Pennsylvania State University College of Medicine, Hershey, PA 17033, USA. [2]Department of Molecular and Precision Medicine, Pennsylvania State University College of Medicine, Hershey, PA 17033, USA. [3]Penn State Cancer Institute, Pennsylvania State University College of Medicine, Hershey, PA 17033, USA. [4]Department of Molecular & Human Genetics, Baylor College of Medicine, Houston, TX 77030, USA. [5]Development, Disease Models & Therapeutics Graduate Program, Baylor College of Medicine, Houston, TX 77030, USA. [6]Department of Integrative Physiology, Baylor College of Medicine, Houston, TX 77030, USA. [7]Program in Developmental Biology, Baylor College of Medicine, Houston, TX 77030, USA. [8]Department of Biostatistics and Data Science, School of Public Health, The University of Texas Health Science Center at Houston, Houston, TX 77030, USA. [9]Children's Medical Center Research Institute, University of Texas Southwestern Medical Center, Dallas, TX 75235, USA. [10]Metabolomics Core Facility, Department of Bioinformatics and Computational Biology, Division of Basic Science Research, The University of Texas MD Anderson Cancer Center, Houston, TX 77030, USA. [11]Department of Molecular Virology and Microbiology, Baylor College of Medicine, Houston, TX 77030, USA. [12]Section of Hematology/Oncology, Department of Pediatrics, Texas Children's Cancer and Hematology Center, Baylor College of Medicine, Houston, TX 77030, USA. [13]Department of Leukemia, The University of Texas MD Anderson Cancer Center, Houston, TX 77030, USA. [14]Department of Genomic Medicine, The University of Texas MD Anderson Cancer Center, Houston, TX 77030, USA. ✉e-mail: xshi@pennstatehealth.psu.edu; nakada@bcm.edu

