## [Transparent Peer Review file · Nature Communications]

Guanine nucleotide biosynthesis blockade impairs MLL complex formation and sensitizes leukemias to menin inhibition

Corresponding Author: Dr Daisuke Nakada

Version 0:

Reviewer comments:

Reviewer #1

(Remarks to the Author)

Xiangguo et al., used liquid chromatography-mass spectrometry to identify metabolic profiles in murine MLL-AF9 driven LSC and bulk leukemia. The authors further assess which specific genes are involved in the maintaining unique metabolism of murine AML. Followed up by identification of therapeutic options with the efficacy in vivo.

My specific concerns: it has been shown by multiple groups that translating findings from murine AML to human AML is not always easy.

Firstly, the authors transition to using human AML cells lines, in which they confirm sensitivity to inhibition of the guanine nucleotide biosynthesis. These results need more clarification. Human MLL-rearranged AML cells lines are not known to have an LSC compartment. It would be fair to say that human AML are homogenous and probably represent blast population, which is supposed to be non-responsive to MPA, MMF, or 6-MP. How would the authors interpret these data?

The authors also use cultured patient derived xenotransplant (PDX) samples in vitro. But these are race to death experiments that are poorly informative since the cells die a little faster or slower but not proliferating.

Overall, it is not simple to fissure out from the text when authors use murine or human AML. The authors need to make changes in the text which will indicate what exact AML sample is being used for which experiments.

Moreover, since the authors have access to MLL-r and MLL-wt PDX sample there should be simple experiments done with either dosing mice bearing MLL-r AML PDX with MPA, MMF, or 6-MP alone or in combination with Menin inhibitors. Without such experiment is impossible to assess the value of whole work.

Reviewer #2

(Remarks to the Author)

In this manuscript, the authors report that the guanine nucleotide biosynthesis pathway is highly activated in leukemia stem cells (LSCs) of MLL-rearranged (MLLr) acute myeloid leukemia (AML). They've demonstrated that suppression of the purine biosynthesis pathway leads to myeloid differentiation of LSCs into mature myeloid cells. They then focused on the guanidine biosynthesis and showed that MLLr AML are among the most sensitive subtype to guanine biosynthesis inhibition. They further showed that inhibition of IMPDH2 suppresses AML initiation and progression by impairing LSC function.

Next, they showed that inhibiting guanine nucleotide biosynthesis is associated with disruption of rRNA transcription and demonstrated that inhibition of Pol I transcription by CX-5461 or BMH-21 induced myeloid differentiation of LSCs,

phenocopying the effects of inhibiting guanine nucleotide biosynthesis.

They identify IMPDH2 activity to be required for maintaining LSCs and that pharmacological IMPDH inhibition reduces the expression, complex formation, and chromatin binding of the LEDGF-menin-MLL complex. Their results also show that disruption of guanine nucleotide or rRNA biosynthesis reduces the expression and complex formation of LEDGF and menin, thereby reducing their chromatin occupancy and target gene expression of the MLL-fusion complex. Finally, they showed that inhibiting guanine nucleotide biosynthesis pathway render MLLr AML cells highly sensitive to menin inhibitors. They postulate that this therapeutic approach may have potential in menin-mutated AML that become resistant to menin inhibitors, but this is not shown.

The findings are novel and the paper is very well written. The experimental design, data analysis, and presentation are well performed. Below is a list of suggestions that can enhance the paper:

As only the effects of MMF on rRNA synthesis were examined, it is important to assess rRNA synthesis rate in Impdh2 Ko cells and potential rescue with the addition of guanosine- to establish that depletion of Impdh2 inhibits Pol I transcription.

The rationale and interpretation for the data in FigS5B and C is confusing. The purpose of the experiment is unclear. CX-5461 treatment leads to increased guanine nucleoside level, so it is not clear why CX-5461 was added with guanosine. This requires clarification.

The statement about inhibition of purine metabolism impairing the translation of menin needs more clarification. From Figure S6L, MMF mainly affects menin expression at the transcriptional level but CX-5461 suppresses menin expression at a translational level.

Figure 2b, requires a better description of the AML LSC+ and LSC- datasets

Line 175 "Upon re-analysis of our previous whole genome CRISPR screen²⁷", description of the screen and its design should be added.

Fig 3G It seems that conditions are mislabelled on the x-axis.

Line 404 "the loss of NF-Y occupancy by MMF" needs rewording to "occupancy at NF-Y motifs" since the data refers to ATAC-seq

Fig 6N define "*" near western blots

Reviewer #3

(Remarks to the Author)

This is an exciting story regarding the role of guanylates in leukemia. The metabolomic and stable isotope tracing studies are well-done, rigorous and correctly interpreted. I have several suggestions that are primarily technical in nature.

GMP is used as both an abbreviation for a metabolite and a cell type. I suggest simplifying the abbreviation scheme.

gGTP levels are shown in Figure 5. My understanding is that this metabolite is often indistinguishable from ATP due to nearly identical molecular weights and retention times. Are the authors certain that they are measuring dGTP and not ATP? Lower labeling in ATP than GTP could just be due to the larger total pool size/concentration of ATP, so the rise in fractional enrichment is slower.

Recommend confirmatory qPCR or western blots to show effective KO of the targets of interest with the CRISPR strategy

Reviewer #4

(Remarks to the Author)

In the manuscript 'Guanine nucleotide biosynthesis blockade impairs MLL complex formation and sensitizes leukemias to menin inhibition' by Xiangguo Shi, et al., the authors aim to prove the role of the guanine nucleotide biosynthesis within MLLr AML. They show that this form of AML has increased purine biosynthesis in mice, which has not been shown in this specific context yet. With the use of a guanine nucleotide biosynthesis inhibitor, MMF, the authors show a role of this metabolic pathway in the function and maintenance of LSCs from MLLr AMLs. Specifically, the authors show that when treating LSCs with MMF it induces AML differentiation (which is mirrored using the RNA pol I Inhibitor CX-546) and leads to a reduction of GMP/GDP. Overall, this study shows a potential interesting target to target AML, especially in the MLLr types, however the authors should perform some additional experiments to strengthen their findings and enhance the therapeutic relevance.

Major comments

1. In figure 1E, the guanosine levels measured are extremely variable. Furthermore, guanosine does not seem to be

- measured at all in AML-bulk even though other members of the purine biosynthetic pathway were measured. This assay should be improved by increasing the sample size and a possible normalization to tackle the variability.
2. In general, the authors use wild-types as controls. However, to my understanding, these cells have not been pre-cultured and the mice have not transplanted. Can the authors comment on this or show (at least in vitro) that empty-vector treated healthy cells display the same differences?
 3. In figure 2A, the authors show that purine biosynthetic genes exhibit a higher expression in LSCs than bulk AML and WBM cells. However, for Pfas and Gmps the variability is again very high. For this assay the sample size should also be increased to determine whether or not the current results are trustworthy.
 4. In both Figure 2A and 2E the authors compare LSCs only to WBM and bulk AML. However, no comparison is made to GMPs (as was done for figure 1). Due to this, it cannot be said for certain that the changes seen here are LSC or GMP specific. The authors should further increase these assays by including GMPs to make sure the results are specific.
 5. In sentence 143/144 the authors state that the increased purine biosynthetic gene expression seen in AML-patient transcriptomic data contribute to the heightened purine biosynthesis in LSCs. However, the authors have not shown that the metabolic changes seen in mice are also found in human AML samples. To state that this gene profile contributes to the heightened biosynthesis the authors should determine the metabolic profile of human AML samples as well.
 6. In sentence 487/488 the authors state that the data shown in figure 7 establishes that there is a synergistical suppression of MLLr AML by inhibition of the purine biosynthesis and menin-MLL interaction. However, it seems only one combinational treatment experiment with 10 samples was performed. Even though this result shows proof of an accumulative effect, it does not necessarily show that it is a synergistic effect as well. Along the same lines (in sentence 493/494) the authors also directly state that the shown inhibition of purine biosynthesis renders AML more sensitive to menin inhibition. As said before, currently they only show an accumulative effect and no synergy. Further exploration of this combinational treatment should be performed to proof that what is shown is more than just an accumulative effect.
 7. The authors show that the guanine nucleotide pathway is important for the maintenance of LSCs in mice and they hypothesize that it could be promising for future treatments in combination with small menin inhibitors. To strengthen this hypothesis the authors should perform a more translational approach to determine whether the effect they are seeing within the mouse is also applicable in a human context. In general, the manuscript would benefit from strengthening the human part (in terms of MLLr and non-MLLr AML, guanosine levels, treatment).
 8. For the translational relevance of their findings, it would be important to know when the mice started receiving the ziftomenib treatment; is it directly after the transplants? After the leukemia has progressed? While overall the findings are interesting, it would be important to know this from a therapeutic angle as well.
 9. The authors should also take into consideration previous publications, Wang et al. (Blood Cancer J. 1:5), which showed that higher levels of Guanosine increase members of the purine biosynthesis which primes AML towards differentiation. How do these results fit the results within this paper? This could for instance be explained by a more detailed exploration of the mechanism driving the effect of guanosine and MMF treatments on AML.
 10. It would be very helpful for the reader to add workflows in the figures and to have more details about the timelines.

Minor comments,

1. In figure 3N/M the authors show that MLLr AMLs are more sensitive to purine biosynthesis inhibition, however certain non-MLLr AMLs also seem to respond. Do the authors have an explanation into what kind of mechanism is used to gain these results? For the authors to state that, they should perform a more mechanistical exploration of the effect of the purine biosynthesis inhibition in MLLr and non-MLLr AMLs to see whether the recorded effect truly is caused by the impairment of formation of the LEDGF/menin complex and its recruitment to the chromatin, as later stated.
2. Sentence 56, it says "... targeted therapeutic strategies to cue this..." spelling mistake.
3. Sentence 146/147, the sentence structure is not correct (one 'of' too many).
4. Sentence 153/154, the authors should specify which species was used for the development of this ChIP-seq dataset.

Version 1:

Reviewer comments:

Reviewer #2

(Remarks to the Author)

The authors have addressed my concerns.

Reviewer #3

(Remarks to the Author)

My comments have been sufficiently addressed

Reviewer #4

(Remarks to the Author)

The authors have revised the manuscript and addressed most of the comments raised. Most importantly, they have added in vivo experiments that enhance the therapeutic relevance, and they have now specified when the data refers to humans or mouse material. Additionally, the concerns raised by Reviewer 1 have also been addressed. I have no further comments regarding the revised manuscript.

Reply to the reviewers' comments:

We would like to thank all four reviewers for their insightful and constructive comments. We have addressed each of them below (our replies are in blue, the reviewers' comments in black).

Reviewer #1 (Remarks to the Author); expert in AML, MLL AML and inhibitors

Xiangguo et al., used liquid chromatography-mass spectrometry to identify metabolic profiles in murine MLL-AF9 driven LSC and bulk leukemia. The authors further assess which specific genes are involved in the maintaining unique metabolism of murine AML. Followed up by identification of therapeutic options with the efficacy *in vivo*.

My specific concerns: it has been shown by multiple groups that translating findings from murine AML to human AML is not always easy.

Response: Thank you for raising this important point regarding the challenges of translating findings from murine models to human AML. We fully acknowledge that the biological differences between species can complicate the direct application of murine AML data to human contexts. We believe that the use of both murine and human AML cells can provide valuable insights into AML mechanisms, allowing us to capture the unique characteristics and complexities of human AML, particularly regarding metabolic and transcriptional regulation pathways. In our study, we have used both murine AML model and human AML cell lines / patient samples to ensure that our findings are more directly relevant to human disease. We appreciate this reviewer's perspective of the importance of validating results in human models to facilitate more accurate therapeutic development.

Firstly, the authors transition to using human AML cell lines, in which they confirm sensitivity to inhibition of the guanine nucleotide biosynthesis. These results need more clarification. Human MLL-rearranged AML cell lines are not known to have an LSC compartment. It would be fair to say that human AML are homogenous and probably represent blast population, which is supposed to be non-responsive to MPA, MMF, or 6-MP. How would the authors interpret these data?

Response: Thank you for the insightful comments. We appreciate the reviewer's emphasis on the homogeneity of human AML cell lines and the typical representation of the blast population. We would like to clarify our findings regarding the sensitivity of human AML cell lines to MMF treatment.

As noted, our data demonstrate that the response to MMF is particularly pronounced in AML cells with MLL rearrangements (Fig. 3d-e and 3m). Our mechanistic studies suggest that this sensitivity is linked to the reliance of these proliferative cells on the oncogenic transcriptional regulation driven by the MLL fusion complex (Figs. 5-6). Disruption of guanosine biosynthesis pathway by MMF disrupts the formation and function of this complex, leading to compromised proliferation and survival. Supporting this, two recent studies (Leukemia. 2022. 36:383-393 and EMBO Mol Med. 2023. 15:e15631) have shown that inhibition of PAICS, an enzyme involved in *de novo* purine biosynthesis, or IMPDH, effectively suppresses the growth of AML cell lines. As pointed out by the reviewer, MLL-rearranged AMLs are atypical in terms of the LSC compartment they harbor; LSC activity of MLLr AML is contained within CD93+ cells that are proliferative (Cell Stem Cell. 17:412-421) and transcriptionally resemble monocytic-LSCs described by Jordan and colleagues (Cancer Discovery. 13:2032-2049) as immunophenotypically and transcriptionally distinct from primitive-LSCs. Our observations indicate that the inhibitory effects of MMF on LSCs are more significant compared to those on bulk AML cells (Fig. 4p). This suggests that LSCs are particularly sensitive to the disruption of the guanosine biosynthesis pathway, consistent with the observation by Jordan and colleagues that monocytic-LSCs depend on purine metabolism.

The authors also use cultured patient derived xenotransplant (PDX) samples in vitro. But these are race to death experiments that are poorly informative since the cells die a little faster or slower but not proliferating.

Response: We thank the reviewer for raising this important point. We fully agree that cultured patient-derived xenotransplant (PDX) samples typically exhibit limited proliferation in vitro. Rather than focusing on the ability of MMF to suppress AML proliferation, we examined the ability of the drug to induce differentiation. After a brief (24 hours) culture, the samples were treated with MMF and differentiation was assessed after 6 days of treatment. Albeit the potential issue with PDX cells not proliferating in these conditions, we observed that MMF is sufficient to induce AML differentiation. We have added this information to the Methods section of the revised manuscript. More importantly, we have generated MLLr PDX models and tested the effects of MMF in vivo, as described below in more detail.

Overall, it is not simple to fissure out from the text when authors use murine or human AML. The authors need to make changes in the text which will indicate what exact AML sample is being used for which experiments.

Response: We appreciate the reviewer's suggestion. To address this, we have provided more detailed information regarding the use of murine and human AML samples in both the main text, figure legend and the supplementary materials. These included lines 155, 160, 168, 188, 190, 210, 223, 250, 340, 396, 407, 786, 819, 878. These clarifications should help distinguish which specific AML samples were used for each experiment.

Moreover, since the authors have access to MLL-r and MLL-wt PDX sample there should be simple experiments done with either dosing mice bearing MLL-r AML PDX with MPA, MMF, or 6-MP alone or in combination with Menin inhibitors. Without such experiment is impossible to assess the value of whole work.

Response: Thanks for the reviewer for this important suggestion. To address this concern, we generated two MLLr PDX leukemia models with samples AML001 and AML013, which carry MLL-AF9 and MLL-AF6, respectively. We treated these PDX models from day 5 post-transplantation with MMF (100 mg/kg/day), a low dose of ziftomenib (12.5 mg/kg/day), or both for 4-5 weeks. We found that MMF treatment significantly reduced the leukemia burden in the bone marrow of mice as a single agent and when co-treated with ziftomenib. These findings have been updated in the revised manuscript (Fig. 7k-l).

Reviewer #2 (Remarks to the Author); expert in in MYC, rRNA synthesis

In this manuscript, the authors report that the guanine nucleotide biosynthesis pathway is highly activated in leukemia stem cells (LSCs) of MLL-rearranged (MLLr) acute myeloid leukemia (AML). They've demonstrated that suppression of the purine biosynthesis pathway leads to myeloid differentiation of LSCs into mature myeloid cells. They then focused on the guanidine biosynthesis and showed that MLLr AML are among the most sensitive subtype to guanine biosynthesis inhibition. They further showed that inhibition of IMPDH2 suppresses AML initiation and progression by impairing LSC function.

Next, they showed that inhibiting guanine nucleotide biosynthesis is associated with disruption of rRNA transcription and demonstrated that inhibition of Pol I transcription by CX-5461 or BMH-21 induced myeloid differentiation of LSCs, phenocopying the effects of inhibiting guanine nucleotide biosynthesis.

They identify IMPDH2 activity to be required for maintaining LSCs and that pharmacological IMPDH inhibition reduces the expression, complex formation, and chromatin binding of the LEDGF-menin-MLL complex. Their results also show that disruption of guanine nucleotide or rRNA biosynthesis reduces the expression and complex formation of LEDGF and menin, thereby reducing their chromatin occupancy and target gene expression of the MLL-fusion complex. Finally, they showed that inhibiting guanine nucleotide biosynthesis

pathway render MLLr AML cells highly sensitive to menin inhibitors. They postulate that this therapeutic approach may have potential in menin-mutated AML that become resistant to menin inhibitors, but this is not shown.

The findings are novel and the paper is very well written. The experimental design, data analysis, and presentation are well performed. Below is a list of suggestions that can enhance the paper:

Response: Thank you for the positive feedback and insightful comments on our manuscript. We appreciate the reviewer's acknowledgement of the novelty of our findings and the quality of the experimental design and data presentation.

As only the effects of MMF on rRNA synthesis were examined, it is important to assess rRNA synthesis rate in *Impdh2* Ko cells and potential rescue with the addition of guanosine- to establish that depletion of *Impdh2* inhibits Pol I transcription.

Response: Thank you to the reviewer for this insightful suggestion. To address this question, we deleted *IMPDH2* in Cas9-expressing MOLM-13 cells and treated them with or without guanosine for 24 hours. Our results show that *IMPDH2* deletion leads to a reduction of rRNA transcription, which was rescued by guanosine supplementation. These findings establish the critical role of guanine biosynthesis in RNA pol I-mediated rRNA transcription. These new data are shown as Fig. S5a in the revised manuscript.

The rationale and interpretation for the data in FigS5B and C is confusing. The purpose of the experiment is unclear. CX-5461 treatment leads to increased guanine nucleoside level, so it is not clear why CX-5461 was added with guanosine. This requires clarification.

Response: Thank you to the reviewer for pointing this out. The purpose of these experiments was to assess whether guanosine supplementation could rescue myeloid differentiation induced by rRNA transcription suppression by CX-5461 and BMH-21. Our initial hypothesis was that guanosine could not rescue myeloid differentiation under these conditions. However, surprisingly, we observed that guanosine treatment partially rescued myeloid differentiation induced by either CX-5461 or BMH-21 (FigS5B and C of the original submission). These data suggest the potential presence of an rRNA-independent mechanism by which guanosine biosynthesis may influence myeloid differentiation, which needs further investigation. It is also unclear if guanosine supplementation affected the ability of CX-5461 and BMH-21 to inhibit RNA polymerase I. In lieu of these limitations, we have removed these data from the revised manuscript.

The statement about inhibition of purine metabolism impairing the translation of menin needs more clarification. From Figure S6L, MMF mainly affects menin expression at the transcriptional level but CX-5461 suppresses menin expression it at translational level.

Response: Thank you for raising this point. Inhibition of purine metabolism affects both the transcriptional and translational regulation of menin and LEDGF, which are encoded by the *Men1* and *Psip1* genes, respectively. To better reflect these findings, we have changed the subtitle of this section to "Inhibition of purine metabolism reduces the expression of LEDGF and menin". We have also edited the concluding sentence of this paragraph to "These findings suggest that LEDGF and menin expression is sensitive to perturbed guanine nucleotide biosynthesis and RNA pol I function".

Figure 2b, requires a better description of the AML LSC+ and LSC- datasets

Response: Thank you for the suggestion. We have provided additional details regarding the AML LSC+ and LSC- datasets as follows in the revised manuscript:

“Additionally, we analyzed publicly available transcriptomics data from AML patients with a range of mutation profiles²³, where functional LSCs have been validated by xenotransplantation, and found that the majority of the purine biosynthetic genes were expressed at higher levels in LSCs compared to non-LSCs (Fig. 2b).”

Line 175 “Upon re-analysis of our previous whole genome CRISPR screen²⁷”, description of the screen and its design should be added.

Response: We have added the screen description and design in the revised manuscript as follows:

“We previously performed a whole genome CRISPR dropout screen in MOLM-13 cells and identified regulators of NAD+ metabolism as AML vulnerabilities²⁷. Upon re-analysis of this screening result...”

Fig 3G It seems that conditions are mislabelled on the x-axis.

Response: We thank the reviewer for pointing out this mislabeling, which we have corrected.

Line 404 “the loss of NF-Y occupancy by MMF” needs rewording to “occupancy at NF-Y motifs” since the data refers to ATAC-seq

Response: Apologies for the mistake. We have changed it accordingly as follows: “Consistent with the possibility that the loss of occupancy at NF-Y motifs abrogates menin/MLL binding to chromatin...”

Fig 6N define “*” near western blots

Response: We have revised the figure legend to indicate that this band is a non-specific reaction due to its presence after immunoprecipitation with a control IgG.

Reviewer #3 (Remarks to the Author); expert in purine/guanine metabolism, metabolomics

This is an exciting story regarding the role of guanylates in leukemia. The metabolomic and stable isotope tracing studies are well-done, rigorous and correctly interpreted. I have several suggestions that are primarily technical in nature.

GMP is used as both an abbreviation for a metabolite and a cell type. I suggest simplifying the abbreviation scheme.

Response: We thank the reviewer for this suggestion. To avoid confusion, we have decided not to abbreviate guanine mononucleotide. Thus, GMP refers to granulocyte-macrophage progenitors. These changes have been made throughout the manuscript.

gGTP levels are shown in Figure 5. My understanding is that this metabolite is often indistinguishable from ATP due to nearly identical molecular weights and retention times. Are the authors certain that they are measuring dGTP and not ATP? Lower labeling in ATP than GTP could just be due to the larger total pool size/concentration of ATP, so the rise in fractional enrichment is slower.

Response: We appreciate this valuable point. To clarify, we have included the ion chromatography mass spectrometry (IC-MS) peaks for both ATP and dGTP (Fig. 1 to reviewer). As shown in the right panel, ATP and dGTP are detected at 37.3 min and 38.5 min, respectively. Regrettably, we had previously mislabeled both ATP and dGTP as dGTP. After reanalyzing the data, we have corrected this labeling in Fig. 5a. However, this correction did not impact our observations and conclusions. We apologize for the oversight.

Recommend confirmatory qPCR or western blots to show effective KO of the targets of interest with the CRISPR strategy

Response: To address this question, we performed immunoblotting for IMPDH2 and MYC protein after deleting the genes using CRISPR/Cas9. These new data are shown as Fig. S2d and S3c in the revised manuscript. We have also validated the effectiveness of our CRISPR strategy by transducing MOLM-13, THP-1 and HL60 cells (with or without Cas9 expression) with a sgRNA construct that expresses sgBFP-mCherry/GFP-2A-BFP lentivirus (Fig. S3b). The results show that expression of sgBFP extinguishes BFP fluorescence, indicating that the Cas9-mediated gene editing is efficient in these cells.

Reviewer #4 (Remarks to the Author); expert in AML, LSC, epigenetics:

In the manuscript 'Guanine nucleotide biosynthesis blockade impairs MLL complex formation and sensitizes leukemias to menin inhibition' by Xiangguo Shi, et al., the authors aim to prove the role of the guanine nucleotide biosynthesis within MLLr AML. They show that this form of AML has increased purine biosynthesis in mice, which has not been shown in this specific context yet. With the use of a guanine nucleotide biosynthesis inhibitor, MMF, the authors show a role of this metabolic pathway in the function and maintenance of LSCs from MLLr AMLs. Specifically, the authors show that when treating LSCs with MMF it induces AML differentiation (which is mirrored using the RNA pol I Inhibitor CX-546) and leads to a reduction of GMP/GDP. Overall, this study shows a potential interesting target to target AML, especially in the MLLr types, however the authors should perform some additional experiments to strengthen their findings and enhance the therapeutic relevance.

Response: Thank you for the positive feedback. As described below, we have performed additional experiments to further the therapeutic relevance of our study.

Major comments

1. In figure 1E, the guanosine levels measured are extremely variable. Furthermore, guanosine does not seem to be measured at all in AML-bulk even though other members of the purine biosynthetic pathway were

measured. This assay should be improved by increasing the sample size and a possible normalization to tackle the variability.

Response: Thank you for pointing this out. We agree that the guanosine levels measured in Figure 1e showed significant variability. To address this, we attempted to increase the sample size by repeating the experiment with six samples each for WBM, GMPs, bulk AML and LSCs populations using mass spectrometry (MS). However, due to recent issues with the quality control of the MS equipment (Fig. 2a to reviewer), we were only able to detect allantoin (Fig. 2b to reviewer), which was found to be elevated in leukemia cells, consistent with our previous data in Fig. 1e showing increased purine metabolites in bulk AML and LSCs. As observed in our experiments, purine metabolites appear to be volatile, making their detection challenging. We have acknowledged this limitation in the revised manuscript as follows:

“Importantly, among all the metabolites enriched in LSCs, we found an increased abundance of metabolites in the purine biosynthetic pathway...(Fig. 1c-e and S1j), although the detection levels of some of these metabolites were variable”.

2. In general, the authors use wild-types as controls. However, to my understanding, these cells have not been pre-cultured and the mice have not transplanted. Can the authors comment on this or show (at least in vitro) that empty-vector treated healthy cells display the same differences?

Response: We would like to clarify the controls used on our experiments pertaining to this question. In Figure 3c where we analyzed the effects of deleting genes involved in purine biosynthesis, the controls are either “input” (day 4 after lentivirus transduction) for time-course experiments, and an sgRNA against ENAM, a gene not required for AML cells, serves as a negative control. As for Figure 3d-l, DMSO (used as a solvent to dissolve MMF, MPA, 6-MP) or sgRNA targeting *Rosa26* was used as a control for the mouse experiment. In the *in vivo* mouse experiments (Figure 4 and 7), we use 0.5% methylcellulose / 0.1% Tween-80 as the control to MMF treatment or PBS as the control to poly (I:C). The MLL-AF9-induced AML cells were derived by infecting LSK cells with a retroviral construct encoding MLL-AF9 followed by transplantation. The AML cells were isolated from such recipient mice once mice become moribund. To address the question of whether non-leukemic (such as the cells infected with an empty vector instead of MLL-AF9) cells are affected by MMF, we have extensively studied the effects of MMF on normal hematopoiesis and hematopoietic stem progenitor cells (HSPCs) (Fig. S4F-M). In brief, we treated C57BL/6 mice with 100 mg/kg/day of MMF for 14 days. We found that MMF treatment only modestly reduce the cellularity of the spleen but not bone marrow or thymus. MMF treatment also had negligible effects on HSPCs. This is in stark contrast to the effects of MMF on AML; the same MMF treatment regimen significantly extended leukemia-free survival and reduced AML burden observed in the blood.

3. In figure 2A, the authors show that purine biosynthetic genes exhibit a higher expression in LSCs then bulk AML and WBM cells. However, for *Pfas* and *Gmps* the variability is again very high. For this assay the sample size should also be increased to determine whether or not the current results are trustworthy.

Response: Thank you for the suggestion. We have isolated LSCs from the murine MLL-AF9 AML model and performed quantitative PCR to compare the gene expression of *Pfas* and *Gmps* to both bulk AML and WBM cells.

[REDACTED]

The results were consistent with our previous findings that *Pfas* and *Gmps* are expressed at higher levels in LSCs. We have updated Fig. 2a with these new data.

4. In both Figure 2A and 2E the authors compare LSCs only to WBM and bulk AML. However, no comparison is made to GMPs (as was done for figure 1). Due to this, it cannot be said for certain that the changes seen here are LSC or GMP specific. The authors should further increase these assays by including GMPs to make sure the results are specific.

Response: Thank you for this suggestion. To address the concern, we performed new experiments comparing the expression of purine biosynthetic genes and *Myc* in WBM and GMPs. We found a slight increase in the expression of some genes, such as *Myc*, *Ppat*, *Gart*, *Ak2*, and *Impdh1/2* in GMPs compared to WBM but the magnitude of the increase was much blunted compared to that we observed when we compared bulk AML or LSCs with normal WBM cells (as in the original submission). These findings suggest that the expression of purine biosynthesis gene is enhanced in AML cells, particularly in LSCs. We have updated Figs. 2a and 2e with these new results.

5. In sentence 143/144 the authors state that the increased purine biosynthetic gene expression seen in AML-patient transcriptomic data contribute to the heightened purine biosynthesis in LSCs. However, the authors have not shown that the metabolic changes seen in mice are also found in human AML samples. To state that this gene profile contributes to the heightened biosynthesis the authors should determine the metabolic profile of human AML samples as well.

Response: We thank the reviewer for this valuable comment. Due to the large number of cells required for metabolic analysis, determining the metabolic profile of human LSCs remains a challenging task and could not be completed within the timeframe of this revision. Instead, we have revised our statement to clarify that "LSCs demonstrate increased purine biosynthetic gene expression" based on transcriptomic data.

6. In sentence 487/488 the authors state that the data shown in figure 7 establishes that there is a synergistical suppression of MLLr AML by inhibition of the purine biosynthesis and menin-MLL interaction. However, it seems only one combinational treatment experiment with 10 samples was performed. Even though this result shows proof of an accumulative effect, it does not necessarily show that it is a synergistic effect as well. Along the same lines (in sentence 493/494) the authors also directly state that the shown inhibition of purine biosynthesis renders AML more sensitive to menin inhibition. As said before, currently they only show an accumulative effect and no synergy. Further exploration of this combinational treatment should be performed to proof that what is shown is more than just an accumulative effect.

Response: We appreciate the reviewer's insightful comment regarding the synergistic effect. We agree that including more experimental groups (≥ 3) for each drug would be necessary to more accurately assess synergism. We also acknowledge the need for further investigation into the effects of MMF plus ziftomenib combination treatment *in vivo* to establish whether the observed effect extends beyond additive effects. Regardless of whether the observed effects of MMF plus ziftomenib treatment are additive or synergistic, we believe that our observations provide an important step in advancing treatments for MLLr AML. As extensively studied in a recent article (Nature Cancer. 4:1693-1704), 95% of FDA approved drug combinations exhibited additive or less than additive effects on progression-free survival and concluded that "synergy is neither a necessary nor common property of clinically effective drug combinations". Since we cannot conclude whether the two drugs had additive or synergistic effects *in vivo*, we have revised our statements as follows.

“... These data establish that the combinatorial inhibition of purine biosynthesis and menin-MLL interaction strongly suppress MLLr AML.”

“... The reduced binding of the LEDGF/menin/MLL-fusion complex and increased binding of myeloid transcription factors were associated with myeloid differentiation of LSCs and enhanced AML sensitivity to menin inhibition.”

7. The authors show that the guanine nucleotide pathway is important for the maintenance of LSCs in mice and they hypothesize that it could be promising for future treatments in combination with small menin inhibitors. To strengthen this hypothesis the authors should perform a more translational approach to determine whether the effect they are seeing within the mouse is also applicable in a human context. In general, the manuscript would benefit from strengthening the human part (in terms of MLLr and non-MLLr AML, guanosine levels, treatment).

Response: We thank the reviewer for this important point. To further the translational relevance of our study, we generated two MLLr PDX leukemia models that carry MLL-AF9 (AML001) or MLL-AF6 (AML013) translocations. We treated these mice from day 5 post-transplantation with MMF (100 mg/kg/day), a low dose of ziftomenib (12.5 mg/kg/day), or both for 4-5 weeks. We found that the addition of MMF significantly reduced the leukemia burden in the bone marrow of mice treated with ziftomenib. These new results are presented in Fig. 7k-l of the revised manuscript.

8. For the translational relevance of their findings, it would be important to know when the mice started receiving the ziftomenib treatment; is it directly after the transplants? After the leukemia has progressed? While overall the findings are interesting, it would be important to know this from a therapeutic angle as well.

Response: Thanks for the reviewer pointing this out. The mice were treated with ziftomenib from day 5 post-transplantation. We have provided the details of the treatment information throughout the revised manuscript.

9. The authors should also take into consideration previous publications, Wang et al. (Blood Cancer J. 1:5), which showed that higher levels of Guanosine increase members of the purine biosynthesis which primes AML towards differentiation. How do these results fit the results within this paper? This could for instance be explained by a more detailed exploration of the mechanism driving the effect of guanosine and MMF treatments on AML.

Response: We were not able to identify Wang et al. in Blood Cancer J 1:5 but found a paper that describes the findings mentioned by the reviewer as Wang et al. (Am J Cancer Res. 2022 Jan 15;12(1):427–444). In this study, Wang et al. treated AML cells with 100 μ M guanosine for 96 hours and found that this treatment induces myeloid differentiation. These findings suggest that the guanosine pathway may drive AML differentiation both when inhibited and overactivated. We have cited and included this paper in Discussion.

10. It would be very helpful for the reader to add workflows in the figures and to have more details about the timelines.

Response: We have revised the data showing survival curves (Fig. 7g and i-j) to indicate the period in which mice were treated and when they were analyzed. We have also included more information in the methods section of the revised manuscript.

Minor comments,

1. In figure 3N/M the authors show that MLLr AMLs are more sensitive to purine biosynthesis inhibition, however certain non-MLLr AMLs also seem to respond. Do the authors have an explanation into what kind of mechanism is used to gain these results? For the authors to state that, they should perform a more mechanistical exploration of the effect of the purine biosynthesis inhibition in MLLr and non-MLLr AMLs to see whether the recorded effect truly is caused by the impairment of formation of the LEDGF/menin complex and its recruitment to the chromatin, as later stated.

Response: Thank you to the reviewer for raising this important point. To investigate the involvement of LEDGF/menin complex in non-MLLr AMLs, such as HL60, upon purine biosynthesis inhibition, we utilized a CRISPR/Cas9 gene editing approach to test the role of *LEDGF* and *MEN1* (which encode LEDGF and menin respectively) in HL60 cells. We performed a competitive growth assay in Cas9-expressing HL-60 cells over a 13-day culture period (Fig. 3 to reviewer). As a positive control, we found that deletion of *MYC* significantly reduced the proportion of sgRNA-expressing cells. In contrast, deletion of *MEN1* or *LEDGF* did not affect fitness, suggesting these genes are not required for HL-60 cell proliferation. In contrast to the crucial role of LEDGF/menin in MLLr AML cells, these results suggest that the LEDGF/menin complex largely dispensable for the proliferation of non-MLLr AML cells, such as HL-60 cells. However, the molecular mechanisms underlying the sensitivity of non-MLL AML cells to purine biosynthesis inhibition require further investigation.

[REDACTED]

2. Sentence 56, it says "... targeted therapeutic strategies to cue this..." spelling mistake.

Response: Thank you for identifying this mistake. This has been corrected to "cure".

3. Sentence 146/147, the sentence structure is not correct (one 'of' too many).

Response: Thank you for pointing this out. The sentence in lines 146/147 has been corrected to:

"...regulates the transcription of key components involved in purine biosynthesis."

4. Sentence 153/154, the authors should specify which species was used for the development of this CHIP-seq dataset.

Response: We have added the information in the revised manuscript as follows:

"An analysis of a murine MLL-AF9 ChIP-seq dataset in MLL-AF9 expressing LSCs24 revealed MLL-AF9 binding at the promoter of Meis1, ..."

"... leveraging publicly available ChIP-seq datasets of murine hematopoietic cells from ChIP-Atlas..."

"Transcription factors binding in the promoter regions (-250bp to +1000bp of start codon) of purine biosynthesis genes in murine hematopoietic cells were downloaded from the ChIP-Atlas dataset..."

We appreciate the reviewers for their constructive comments and for determining that the manuscript is now publishable. Below are the reviewers' comments in normal font and our responses in italics.

REVIEWERS' COMMENTS

Reviewer #2 (Remarks to the Author):

The authors have addressed my concerns.

Response: We thank the reviewer for the constructive comments.

Reviewer #3 (Remarks to the Author):

My comments have been sufficiently addressed

Response: We thank the reviewer for the constructive comments.

Reviewer #4 (Remarks to the Author):

The authors have revised the manuscript and addressed most of the comments raised. Most importantly, they have added in vivo experiments that enhance the therapeutic relevance, and they have now specified when the data refers to humans or mouse material. Additionally, the concerns raised by Reviewer 1 have also been addressed. I have no further comments regarding the revised manuscript.

Response: We thank the reviewer for the constructive comments and assessing our response to the other reviewer(s).